# Understanding Self-Supervised Learning via Latent Distribution Matching

**Fabian A Mikulasch** [1]    **Friedemann Zenke** [1 2]

## Abstract

Self-supervised learning (SSL) excels at finding general-purpose latent representations from complex data, yet lacks a unifying theoretical framework that explains the diverse existing methods and guides the design of new ones. We cast SSL as latent distribution matching (LDM): learning representations that maximize their log-probability under an assumed latent model (alignment), while maximizing latent entropy to prevent collapse (uniformity). This view unifies independent component analysis with contrastive, non-contrastive, and predictive SSL methods, including stop gradient (stopgrad) approaches. Leveraging LDM, we derive a nonlinear, sampling-free Bayesian filtering model with a Kalman-based predictor for high-dimensional timeseries. We further prove that predictive LDM yields identifiable latent representations under mild assumptions, even with nonlinear predictors. Overall, LDM clarifies the assumptions behind established SSL methods and provides principled guidance for developing new approaches.

## 1. Introduction

Self-supervised learning (SSL) has become a central paradigm for representation learning, enabling models to extract useful structure from unlabeled data across vision, language, and audio domains (Noroozi & Favaro, 2016; Oord et al., 2018; Dawid & LeCun, 2024; Gui et al., 2024). By learning from relationships between multiple views of the same data, such as augmentations, temporal neighbors, or masked context, modern SSL methods often provide more useful representations for downstream tasks than traditional likelihood-based approaches like autoencoders, which

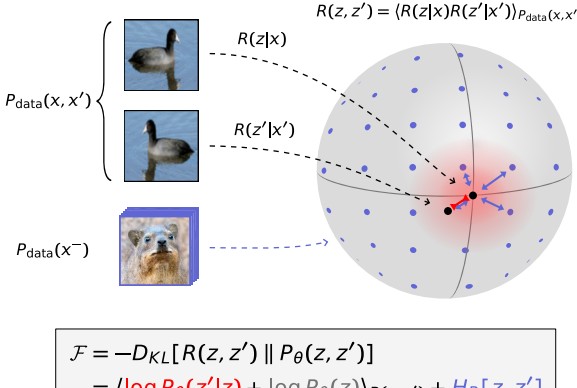

$$R(z, z') = \langle R(z|x)R(z'|x')\rangle_{P_{\text{data}}(x,x')}$$

$$\mathcal{F} = -D_{KL}[R(z, z') \,\|\, P_\theta(z, z')]$$
$$= \langle \log P_\theta(z'|z) + \log P_\theta(z)\rangle_{R(z,z')} + H_R[z, z']$$

*Figure 1.* We formulate SSL as a distribution matching problem in which the transformed data distribution $R(z, z')$ is matched to the latent model $P_\theta(z, z')$. The transformation is deterministic $R(z|x) = \delta(z - f(x))$, where $f(x)$ is a deep network. The model likelihood $\log P_\theta$ and latent entropy $H_R$ correspond to alignment and uniformity terms in the loss function (Wang & Isola, 2020).

emphasize low-level reconstruction and can overfit to superficial details.

Among SSL approaches, latent predictive models have recently achieved state-of-the-art performance on temporal and sequential data (Oord et al., 2018; Bardes et al., 2024; Assran et al., 2025). Here, instead of predicting upcoming data points as in temporal generative models, the goal is to predict future latent representations, while preventing representational collapse. Despite the empirical success of predictive SSL, these models typically rely on heuristic objectives or regularization strategies, while our understanding of *why* they work, and how to design them systematically, remains incomplete (Reizinger et al., 2025).

A common way to interpret SSL is through a geometric alignment perspective (Wang & Isola, 2020): representations of paired views are encouraged to be close in latent space, while additional "uniformity" or repulsion terms in the loss prevent representational collapse. Although this view offers intuition and has influenced practical designs, it does not constitute a formal statistical foundation for SSL. In particular, it provides limited guidance for models that do not explicitly rely on contrastive or repulsive terms, such as BYOL (Grill et al., 2020) or SimSiam (Chen et al., 2020).

[1]Friedrich Miescher Institute for Biomedical Research, 4056 Basel, Switzerland [2]Faculty of Science, University of Basel, 4033 Basel, Switzerland. Correspondence to: Fabian A Mikulasch <fabian.mikulasch+ldm@fmi.ch>, Friedemann Zenke <friedemann.zenke+ldm@fmi.ch>.

*Proceedings of the 43rd International Conference on Machine Learning*, Seoul, South Korea. PMLR 306, 2026. Copyright 2026 by the author(s).

Another influential line of work frames SSL as mutual information (MI) maximization between paired views to preserve shared information while discarding noise (Oord et al., 2018; Tian et al., 2020; Shwartz-Ziv et al., 2023; Gálvez et al., 2023; Shwartz Ziv & LeCun, 2024). While appealing, MI-based interpretations face a fundamental limitation: MI is invariant under arbitrary invertible transformations $\phi$ and $\psi$, i.e.,

$$I[x, y] = I[\phi(x), \psi(y)] \tag{1}$$

and, therefore, does not by itself promote semantically meaningful or identifiable representations. Empirically, MI maximization has proven neither necessary nor sufficient for successful SSL (Tschannen et al., 2020), leaving its precise role elusive.

In contrast, formal guarantees for successful representations in SSL stem from the latent distribution matching (LDM) principle (Zimmermann et al., 2021; Khemakhem et al., 2020b). If a learned latent distribution matches the distribution of the "true" underlying latent variables, then—under assumptions that depend on the model—the representations can recover these variables up to trivial equivalences, such as permutations or affine transformations.

Motivated by these results, we ask whether LDM can serve as a unifying objective for SSL rather than an implicit or auxiliary principle. Our main contributions are:

**A unified distribution-matching framework for SSL.** We formulate SSL as matching a data distribution to an assumed latent model (Fig. 1) and show how a wide range of existing SSL algorithms can be recovered as instances of LDM (Table 1), with differences between methods arising naturally from distinct choices of latent models or entropy estimators.

**Clarifying the role of MI maximization.** We show that in LDM, MI maximization contributes little to representation quality beyond entropy maximization because it is invariant to arbitrary invertible transformations. Instead, approximate MI maximization implicitly performs distribution matching.

**Derivation of new SSL objectives.** We show how LDM naturally leads to new SSL variants with beneficial properties. As an example, we derive a predictive SSL model with Kalman-based latent dynamics combined with a deep encoder, thereby enabling principled uncertainty quantification over latent states.

**Identifiability guarantees for predictive SSL.** We provide identifiability results for predictive SSL models trained via LDM, showing that they can recover underlying latent variables up to affine transformations, thereby providing a formal explanation for their strong empirical performance.

## 2. Prior work

We organize the prior work according to theoretical approaches for understanding SSL, notable identifiability results on SSL, and previous use of LDM in SSL.

**Theoretical perspective on SSL.** There has been significant interest in understanding SSL through theoretical analysis (e.g., Saunshi et al., 2019; Wang & Isola, 2020; Ben-Shaul et al., 2023). A line of work closely related to this article aims to connect contrastive SSL (e.g., CPC) to latent variable models (Von Kügelgen et al., 2021; Zimmermann et al., 2021; Aitchison & Ganev, 2024; Nakamura et al., 2023; Bizeul et al., 2024). Others proposed information-theoretic interpretations of regularization-based SSL (e.g., VICReg) (Shwartz-Ziv et al., 2023) and clustering/contrastive SSL (Gálvez et al., 2023). Finally, stopgrad-based SSL was analyzed using several approaches (Tian et al., 2021; Halvagal et al., 2023; Nakamura et al., 2023), yet a unified treatment is lacking.

**Identifiability in SSL and ICA.** Identifiability is a core concept in linear independent component analysis (ICA) (Hyvärinen & Oja, 1999), which guarantees the recovery of "true" underlying variables up to trivial transformations. The concept was later extended to nonlinear ICA using assumptions such as temporal structure or auxiliary variables (Sprekeler et al., 2014; Khemakhem et al., 2020a; Hyvarinen et al., 2019; Roeder et al., 2021). More recently, these insights also enabled proving identifiability for SSL models (Von Kügelgen et al., 2021; Zimmermann et al., 2021; Daunhawer et al., 2023; Reizinger et al., 2024; Laiz et al., 2025). However, most of these results were derived for specific algorithms, emphasizing the need for a more general framework.

**Distribution matching approaches in SSL.** Zimmermann et al. (2021) implicitly showed that contrastive SSL can be understood as conditional LDM via the reverse KL-Divergence (Appendix A.2.1). Other LDM approaches to SSL have been proposed from the perspective of optimal transport (Chen et al., 2024; Jiao et al., 2025) or rate reduction (Ma et al., 2022). Balestriero & LeCun (2025) proposed single-variable LDM based on an isotropic Gaussian as basis for SSL. In this nonlinear setting, single-variable LDM does not provide identifiability guarantees and must be augmented with additional loss terms. Still, how one can use LDM to design identifiable SSL algorithms has not been explored.

## 3. Distribution matching framework

To motivate the latent distribution matching (LDM) framework, we first revisit the maximum likelihood objective,

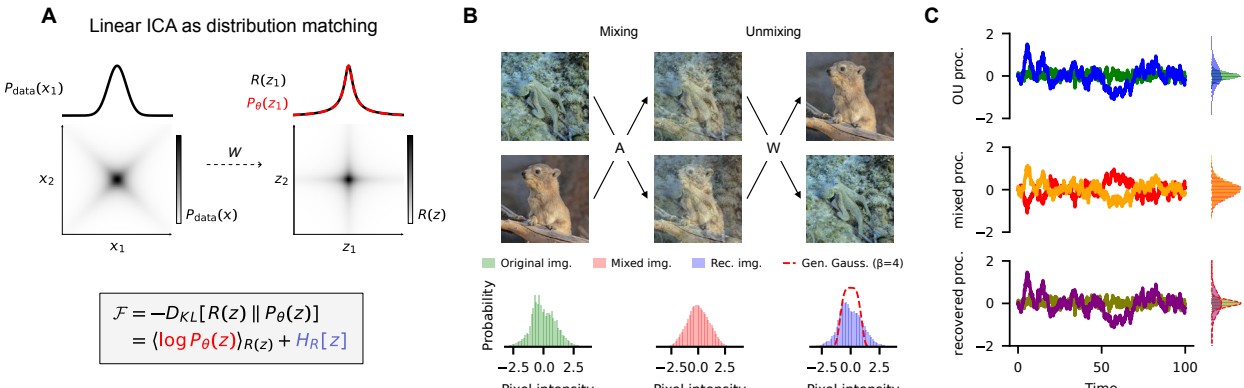

*Figure 2.* Source recovery with LDM in linear ICA. **A** Linear ICA assumes that the data distribution has independent factors, that can be recovered by aligning them with the correct underlying independent distribution (Cardoso, 2002). **B** Distributions of pixel intensities in natural images are non-Gaussian (Hyvärinen & Oja, 1999). In contrast, mixed images are closer to Gaussian, as expected from the central limit theorem. Disentanglement proceeds by learning $W$ to recover an assumed short-tailed distribution (red dashed line). **C** Also Gaussian sources can be disentangled, which, however, requires more assumptions on the data generating process. Here we recover the outputs of two Ornstein-Uhlenbeck processes assuming known variances and that $W$ is volume preserving (determinant of $|W| = 1$).

which is also used, e.g., in linear ICA (Hyvärinen & Oja, 1999) or normalizing flow networks (Papamakarios et al., 2021). In these approaches, models often compute the data-likelihood in latent, instead of data space, which is possible for an invertible data generation process $g \equiv f^{-1}$ (see Appendix A.4.1 and Papamakarios et al., 2021; Cardoso, 2002, see Appendix A.1 for a summary of our notation)

$$\langle \log P_\theta(x) \rangle_{P_{\text{data}}(x)} \propto \underbrace{\langle \log P_\theta(f(x)) \rangle_{P_{\text{data}}(x)}}_{\text{Likelihood}} + \underbrace{H_{P_{\text{data}}}[f(x)]}_{\text{Entropy}}$$
$$= -D_{KL}[P_{\text{data}}(f(x)) \parallel P_\theta(f(x))] , \quad (2)$$

where $P_\theta$ is the model distribution with parameters $\theta$. Importantly, this does not require the encoder $f$ to be invertible in general but only *on the data manifold* (Appendix A.4.1), which is the same assumption underlying the identifiability results for SSL (Section 7). Thus, for an encoder that is invertible on the data manifold, distribution matching in latent space is equivalent to maximum likelihood learning.

### 3.1. Linear ICA as latent distribution matching

To build intuition, we briefly review how linear ICA can be understood as LDM (see Choi et al., 2005; Hyvärinen & Oja, 1999, for more thorough reviews on ICA). Several linear ICA variants, including InfoMax and likelihood-based methods, assume that the data can be disentangled with an unmixing matrix $W$ to match an assumed latent distribution $P_\theta(z) = \prod_i P_\theta(z_i)$ that factorizes over latent factors (Hyvärinen & Oja, 1999; Cardoso, 2002). This unmixing corresponds to an LDM goal (Fig. 2A). If the data is linearly transformed as $z = Wx$, the transformed data distribution is $R(z) = \langle \delta(z - Wx) \rangle_{P_{\text{data}}(x)}$, which is matched to $P_\theta(z)$

via (Cardoso, 2002)

$$\mathcal{F}_{\text{ICA}} = -D_{KL}[R(z) \parallel P_\theta(z)] = \langle \log P_\theta(z) \rangle_{R(z)} + H_R[z]$$
$$= \langle \log P_\theta(z) \rangle_{R(z)} + \frac{1}{2} \log |WW^T| + H_{P_{\text{data}}}[x] , \quad (3)$$

where we used the formula for the change of variables in the entropy. Since the data entropy term is constant it can be omitted, and maximizing the likelihood and log-determinant terms leads to disentangled factors (Fig. 2).

The assumption of known source distributions might seem limiting. However, on the one hand, the assumed latent distribution need not match precisely for successful recovery; it is typically sufficient if the general shape matches (e.g., sub- or super-gaussian, Hyvärinen & Oja, 1999). On the other hand, since we have a well-defined log-likelihood function, we can optimize the distribution parameters $\theta$ to better fit the data (Hyvärinen & Oja, 1999). This last insight will be especially relevant to understand predictive SSL (Section 6).

## 4. SSL from latent distribution matching

The central limitation of single-variable ICA, as formulated above, is that, for a nonlinear unmixing function $z = f(x)$, the problem becomes ill-defined (Hyvärinen et al., 2023). However, it is possible to formulate a well-defined source recovery problem by specifying additional conditional dependencies in the latent variable distribution (Sprekeler et al., 2014; Hyvarinen et al., 2019; Zimmermann et al., 2021; Khemakhem et al., 2020a; Hyvärinen et al., 2023). The basic intuition is that these additional conditional constraints in latent space provide enough information to determine a unique solution to the nonlinear unmixing problem, i.e., up

to trivial transformations.

Inspired by this we define a *joint distribution* matching goal (cf. Fig. 1; general formulation in Appendix A.3). We first define the reshaped latent distribution via a recognition density $R(z|x)$ and the data distribution

$$R(z, z') := \langle R(z|x)R(z'|x') \rangle_{P_{\text{data}}(x,x')} \ . \quad (4)$$

In the following, we assume a deterministic recognition function $R(z|x) = \delta(z - f(x))$, where $f(x)$ is parametrized by a deep neural network. We explore nondeterministic encoders in Appendix A.8. With these definitions, we can formulate the LDM SSL goal function as follows

$$\mathcal{F}_{\text{LDM}} = -D_{KL}[R(z, z') \parallel P_\theta(z, z')]$$
$$= \underbrace{\langle \log P_\theta(z, z') \rangle_{R(z,z')}}_{\text{Alignment}} + \underbrace{H_R[z, z']}_{\text{Uniformity}} \ . \quad (5)$$

Two observations are worth highlighting. First, the likelihood term promotes alignment, whereas the entropy term rewards uniformity (cf. Fig. 1), as discussed previously by Wang & Isola (2020). Importantly, the uniformity term encourages invertibility of the encoder on the data manifold, by preventing different data points from collapsing to the same point in latent space (Appendix A.4.2). Second, a well-known property of the KL divergence is that it is nonnegative, and zero only if the distributions match. Hence, if $\mathcal{F}_{\text{LDM}}$ is maximized and $R$ and $P_\theta$ have the same support, $R(z, z') = P_\theta(z, z')$, and in particular $R(z|z') = P_\theta(z|z')$. As outlined above, this conditional LDM has been shown to recover the underlying latent variables up to trivial transformations when the data-generating process is invertible (Hyvarinen et al., 2019; Zimmermann et al., 2021; Roeder et al., 2021). We will use this property for proving the theorems in Section 7.

### 4.1. Distribution matching and MI maximization

LDM and MI maximization are closely related, which can be shown through a goal function previously proposed by Aitchison & Ganev (2024). Specifically, based on variational inference principles, they arrived at

$$\mathcal{F}_{\text{MI}} = -D_{KL}[R(z, z') \parallel P_\theta(z, z')] + I_R[z, z']$$
$$= \langle \log P_\theta(z, z') \rangle_{R(z,z')} + 2H_R[z] \ , \quad (6)$$

which differs from $\mathcal{F}_{\text{LDM}}$ only through the MI term. For the second line we assumed that $R(z)$ equals $R(z')$ and thus $H_R[z] = H_R[z']$ to simplify notation. This expression constitutes a well-known variational bound on the MI of $z$ and $z'$ (Barber & Agakov, 2004; Poole et al., 2019, Appendix A.2.3). To see this property, note that Equation (6) equals $I_R[z, z']$ if $P_\theta(z, z')$ is learned to match $R(z, z')$.[1]

[1]This relation of SSL to MI maximization is only partial, since typically $P_\theta(z, z')$ is restricted, or not learned at all. For an extended discussion see Tschannen et al. (2020).

The SSL formulation by *Aitchison et al.* is therefore more closely related to many previous SSL approaches, which are often motivated through MI maximization (Shwartz-Ziv et al., 2023; Oord et al., 2018).

The key link between $\mathcal{F}_{\text{LDM}}$ and $\mathcal{F}_{\text{MI}}$ is that, for an encoder that is invertible on the data manifold, MI is already maximized (Appendix A.4.3). Because entropy maximization encourages invertibility, we expect the additional MI maximization in Equation (6) to contribute little to the resulting latent representation. In the following, we show how popular SSL algorithms can be derived from $\mathcal{F}_{\text{MI}}$ and then demonstrate in simulations that they yield virtually identical representations to those of matching SSL algorithms derived from $\mathcal{F}_{\text{LDM}}$. To this end, we first discuss briefly how different entropy estimators relate to different SSL approaches.

### 4.2. Entropy estimation

While for categorical latent distributions it is sometimes possible to compute the entropy directly, for continuous variables it has to be approximated.[2] This problem is challenging and has no unique best solution (Paninski, 2003). However, empirically, we find that relatively basic estimators yield good results as long as they are robust and efficiently optimized.

Depending on the approximation, it is possible to recover contrastive, non-contrastive, or other SSL approaches (for details, see Appendix A.5). Entropy estimation based on kernel density estimation (KDE) leads to contrastive SSL such as SimCLR. In contrast, parametric estimation can be related to non-contrastive SSL such as VICReg. We further show that conditional entropy estimation with a predictor is implicitly implemented by SSL models with a predictor and stop gradient (stopgrad) (Section 6). In addition, we propose that entropy estimation for SSL can be implemented through a K-nearest neighbors (kNN) estimator, which is closely related to the contrastive approach.

## 5. Contrastive and non-contrastive learning

**VICReg** is a popular non-contrastive SSL approach that leverages variance and covariance regularization of latent representations (Bardes et al., 2021; Halvagal & Zenke, 2023). To derive a similar algorithm we make the choices

$$P_\theta(z) = Flat,$$
$$P_\theta(z'|z) = \mathcal{N}(z'; \mu = z, \Sigma = \sigma^2 I) \ . \quad (7)$$

[2]Note that there is no difference in principle between estimating the entropy of a single variable or joint distribution ($H_R[z]$ and $H_R[z, z']$, respectively), since the latter can be transformed into the former simply by concatenating the two variables. We therefore do not treat them separately here.

*Table 1.* Summary of model mapping to the LDM framework. *MI max.* denotes whether $\mathcal{F}_{\text{LDM}}$ ($\circ$) or $\mathcal{F}_{\text{MI}}$ ($\bullet$) is employed.

| SSL Method | Latent distribution $P_\theta$ (prior + cond.) | Entropy estimator | MI max. |
|---|---|---|---|
| VICReg (Bardes et al., 2021) | flat + cond. Gaussian | parametric (Gaussian entropy) | $\bullet$ |
| SimCLR (Chen et al., 2020) | uniform + cond. von Mises-Fisher | kernel density | $\bullet$ |
| CPC (Oord et al., 2018) | empirical + predictive cond. von Mises-Fisher | see Aitchison & Ganev (2024) | $\bullet$ |
| BYOL/SimSiam (Grill et al., 2020; Chen & He, 2021) | empirical + cond. von Mises-Fisher | conditional entropy plugin | $\circ$ |
| JEPA (e.g., Bardes et al., 2024; Mohammadi et al., 2025) | predictive cond. Gaussian | conditional entropy plugin | $\circ$ |

This model assumes that two related observations $z$ and $z'$ are much closer than expected under the improper flat prior. The goal function of VICReg can be derived from $\mathcal{F}_{\text{MI}}$ by employing the parametric entropy estimator under a normal distribution (Appendix A.6.1)

$$\mathcal{F}_{\text{MI}} = -\frac{1}{2\sigma^2} \left\langle \|f(x) - f(x')\|^2 \right\rangle_{P_{\text{data}}(x,x')} + \log |\Sigma_z| \, , \tag{8}$$

where $\Sigma_z$ is the covariance matrix of $z$, and $\sigma^2$ trades off between the push and pull terms. As discussed by Shwartz-Ziv et al. (2023), the variance-covariance regularization of VICReg can be understood to approximately maximize the single-variable covariance log-determinant $\log |\Sigma_z|$. This relation can be seen by performing a Taylor expansion of the log-determinant $\log |\Sigma_z| \approx \sum_i \log \Sigma_{ii} - \frac{1}{2} \sum_{j \neq i} \frac{\Sigma_{ij} \Sigma_{ji}}{\Sigma_{ii} \Sigma_{jj}}$, which is similar to the regularizer in VICReg. Thus, VICReg can be understood as LDM to a conditional normal distribution, with additional MI maximization between latent representations.

If, instead, we employ the proposed goal function $\mathcal{F}_{\text{LDM}}$, the following goal function can be derived:

$$\mathcal{F}_{\text{LDM}} = -\frac{1}{2\sigma^2} \left\langle \|f(x) - f(x')\|^2 \right\rangle_{P_{\text{data}}(x,x')} + \frac{1}{2} \log |\Sigma_{(z,z')}| \, , \tag{9}$$

where $\Sigma_{(z,z')}$ is the covariance matrix of the concatenated vector $(z, z')$.

**SimCLR** is a simple contrastive learning algorithm, in which latents $z$ are lying on the unit sphere (Chen et al., 2020). To derive it, we assume

$$P_\theta(z) = \frac{1}{Z},$$
$$P_\theta(z'|z) = \frac{1}{Z} \exp(\beta z^T z') \, . \tag{10}$$

Here, $Z$ is the normalization from integrating over the sphere, and related latents are assumed to be locally close based on the von Mises-Fisher (vMF) distribution, i.e., the spherical normal distribution. If the entropy is approximated via KDE with bandwidth $1/\beta$ then based on $\mathcal{F}_{\text{MI}}$ we recover

the original goal function (Appendix A.6.1)

$$\mathcal{F}_{\text{MI}} = \left\langle \beta f(x)^T f(x') \right\rangle_{P_{\text{data}}(x,x')} - 2 \left\langle \log \langle \exp\{\beta f(x)^T f(x^-)\} \rangle_{P_{\text{data}}(x^-)} \right\rangle_{P_{\text{data}}(x)} \, . \tag{11}$$

Thus, SimCLR can be understood as LDM to a conditional vMF distribution with added MI maximization.

### 5.1. Empirical comparison

To compare both SSL formulations via $\mathcal{F}_{\text{LDM}}$ and $\mathcal{F}_{\text{MI}}$, we tested how MI maximization and the choice of entropy estimator affect representation learning on natural image datasets. We implemented SSL algorithms based on all combinations of assumed latent space (sphere or plane), entropy estimators (KDE, kNN, parametric with LogDet), and whether MI is maximized ($\mathcal{F}_{\text{MI}}$) or not ($\mathcal{F}_{\text{LDM}}$). Note that although LogDet technically is not a valid entropy estimator for spherical distributions, we included it here for completeness. We then assessed representational quality using linear probing (Table 2). We found no systematic performance difference between models with and without MI maximization, as also reflected in network gradients (Appendix Fig. A2). Instead, the combination of latent space and entropy estimator had a clear and systematic impact on validation performance. Also, when analyzing the dimensionality of latent representations, we found no apparent difference between $\mathcal{F}_{\text{MI}}$ and $\mathcal{F}_{\text{LDM}}$ (Fig. 3). Instead, the assumed latent space, and especially the entropy estimator, had a more pronounced effect.

## 6. Predictive learning of dynamical systems

An appealing feature of the LDM framework is that it also allows us to treat temporal models. To this end, we first generalize the goal function to the temporal case

$$\mathcal{F}_{\text{LDM}} = -D_{KL}[R(\boldsymbol{z}) \| P_\theta(\boldsymbol{z})]$$
$$= \sum_t \langle \log P_\theta(z_t|z_{:t}) \rangle_{R(z_t, z_{:t})} + H_R[z_t|z_{:t}] \, , \tag{12}$$

where $z_{:t} = \{z_s : 0 \leq s < t\}$. For additional MI maximization we find

$$\mathcal{F}_{\text{MI}} = -D_{KL}[R(\boldsymbol{z}) \| P_\theta(\boldsymbol{z})] + \sum_t I_R[z_t, z_{:t}]$$
$$= \sum_t \langle \log P_\theta(z_t|z_{:t}) \rangle_{R(z_t, z_{:t})} + H_R[z_t] \, . \tag{13}$$

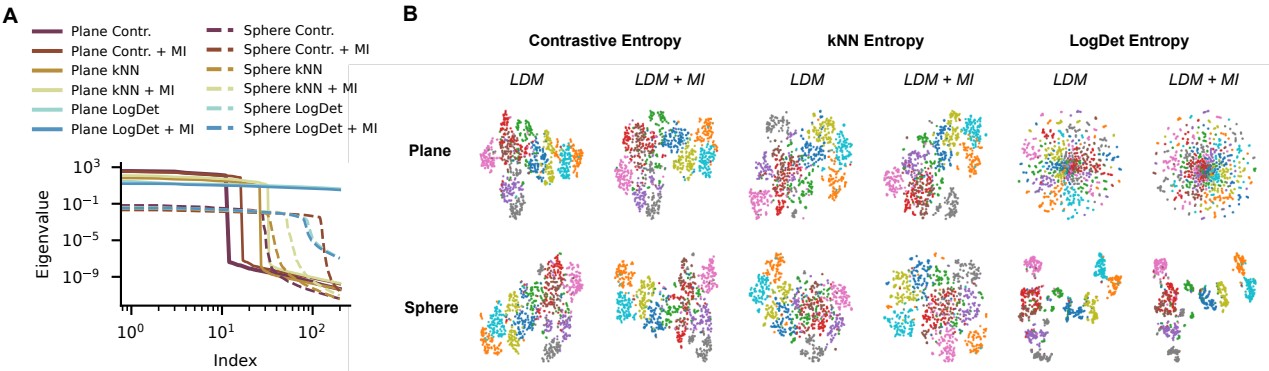

*Figure 3.* Comparison of learned image representations on CIFAR-10. **A** The eigenspectrum of the learned representations generally decays more slowly for parametric entropy estimators, both on the plane (solid) and the sphere (dashed). Whether or not MI was maximized (+ MI) had little impact on the spectrum. The observed cutoff at low double digits is consistent with previous estimates of intrinsic dimensionality of CIFAR-10 (Pope et al., 2021). **B** T-SNE embeddings (Maaten & Hinton, 2008) of representations paint a similar picture in which MI maximization has little impact. Color denotes label.

*Table 2.* Validation accuracy of distinct SSL variants for linear probing on different datasets. We used a standard SSL image pretraining setup as described by da Costa et al. (2022). *MI* denotes whether $\mathcal{F}_{\text{LDM}}$ (○) or $\mathcal{F}_{\text{MI}}$ (●) was used. Plane-LogDet-● corresponds to VICReg, Sphere-Contr.-● corresponds to SimCLR. SimCLR can also be derived via the reverse KL divergence (Zimmermann et al., 2021), potentially explaining the difference in performance. See Table A1 for extended results, including SVHN.

| Space | Model Entropy est. | MI | CIFAR-10 Top-1 Acc. | CIFAR-100 Top-1 Acc. | Imagenet-100 Top-1 Acc. |
|---|---|---|---|---|---|
| Plane | Contr. | ○ | 91.9±0.1 | 65.3±0.3 | 72.6 |
| Plane | Contr. | ● | 92.1±0.1 | 65.3±0.2 | 72.7 |
| Plane | kNN | ○ | 92.1±0.2 | 65.6±0.4 | 74.3 |
| Plane | kNN | ● | 91.9±0.1 | 65.8±0.4 | 73.7 |
| Plane | LogDet | ○ | 92.1±0.1 | **69.5**±0.2 | **75.9** |
| Plane | LogDet | ● | 91.9±0.1 | 68.6±0.1 | 74.7 |
| Sphere | Contr. | ○ | 90.9±0.1 | 64.8±0.2 | 72.1 |
| Sphere | Contr. | ● | 91.4±0.2 | 66.0±0.2 | 73.1 |
| Sphere | kNN | ○ | 90.2±0.0 | 64.3±0.1 | 72.6 |
| Sphere | kNN | ● | 90.0±0.2 | 64.5±0.2 | 73.3 |
| Sphere | LogDet | ○ | 91.4±0.2 | 65.4±0.2 | 73.0 |
| Sphere | LogDet | ● | 91.2±0.1 | 65.4±0.2 | 72.3 |

Depending on the choice of predictor $P_\theta(z_t|z_{:t})$ and entropy estimator, we can relate $\mathcal{F}_{\text{LDM}}$ and $\mathcal{F}_{\text{MI}}$ to previously proposed models. The key difference from the image models in Section 5 is that these temporal models also optimize the latent models' parameters $\theta$.

**Predictive SSL with stopgrad.** The LDM goal $\mathcal{F}_{\text{LDM}}$ requires to either maximize the total entropy $H_R[\boldsymbol{z}]$ or the conditional entropies $H_R[z_t|z_{:t}]$. For long time series, the latter is much easier to achieve, since $\boldsymbol{z}$ becomes high-dimensional. In fact, predictive SSL approaches with stopgrad (Grill et al., 2020; Bardes et al., 2024; Mohammadi et al., 2025) implicitly implement approximate conditional entropy maximization, which we show in Appendix A.6.2. These approaches approximate the goal function as

$$\mathcal{F}_{\text{LDM}} \approx \sum_t \langle \log P_\theta(SG[z_t]|z_{:t}) \rangle_{R(z_t, z_{:t})} . \quad (14)$$

In particular, if the predictor $P_\theta(z_t|z_{:t})$ is a conditional Gaussian distribution with mean $p(z_{:t})$, we recover the loss functions of previous predictive SSL approaches as proportional to $\|SG[z_t] - p(z_{:t})\|^2$.

**Contrastive predictive coding (CPC).** For $\mathcal{F}_{\text{MI}}$, single-timestep entropy can easily be maximized through either KDE, kNN, or parametric estimators. Note that InfoNCE has been derived before from $\mathcal{F}_{\text{MI}}$ under a slightly different model (Aitchison & Ganev, 2024).

### 6.1. Nonlinear Bayesian filtering with Kalman predictor

To demonstrate how LDM enables deriving new SSL algorithms, we now consider a model in which the predictor is given by a Kalman filter. This enables to analytically quantify uncertainty in latent representations in a nonlinear dynamical system with linear latent dynamics. We summarize the assumptions in Fig. 4B, which lead us to the following model log-likelihood:

$$\log P_\theta(z_t|z_{:t}) \propto -\frac{1}{2} e_t^\top \Sigma_e^{-1} e_t - \frac{1}{2} \log |\Sigma_e| , \quad (15)$$
$$e_t = z_t - DAh_{t-1} ,$$

where $h_t$ is the estimated hidden latent state of the filter and $\Sigma_e$ the estimated prediction error covariance. $h_t$ and $\Sigma_e$ can be computed analytically from past latent states $z_t$ through the well-known Kalman filter equations.

To understand the differences between representation learning with CPC and predictive stopgrad, we devised a nonlinear filtering problem based on synthetic videos. Since we employed a Kalman-based latent state predictor, we generated linear hidden dynamics that were then nonlinearly transformed into image space (Fig. 4A, Appendix Fig. A3). After learning, both CPC (with MI maximization) and predictive stopgrad SSL (without MI maximization) recovered

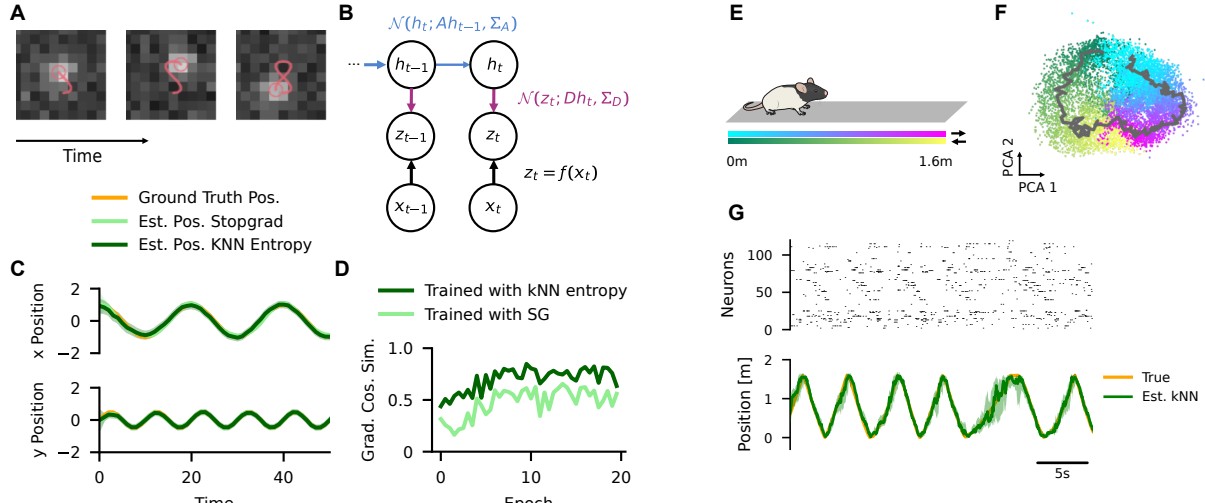

*Figure 4.* Predictive distribution matching in latent space using a nonlinear Bayesian filtering model with Kalman-based predictor. **A** Example frames of synthetic dataset of a high dimensional noisy observable with linear latent dynamics. The red line denotes ground truth position. See Appendix, Fig. A3 for a more nonlinear task. **B** We use a Kalman filter backbone for the predictor $P_\theta(z_t|z_{:t})$ with hidden states $h_t$ and latent "observations" $z_t$. **C** Estimated position and ground truth over time. After learning, we can linearly decode the true position and uncertainty from the Kalman hidden states $h_t$. Training with MI maximization (kNN entropy estimator) or without (stopgrad) results in approximately equal performance. **D** The cosine similarity of gradients of network weights w.r.t. entropy estimators also shows that both approaches lead to similar optimization. Similarity increases with training when the predictor becomes more accurate and so does the stopgrad entropy estimator. **E** Experimental setup in Grosmark & Buzsáki (2016). **F** The learned latent states $h_t$ are arranged in a circle according to the position and direction of the rat. One example trajectory over time is displayed in gray. **G** Example neural spiking timeseries and the estimated position based on the model. Shaded areas denote 95% confidence intervals based on model covariances $\Sigma_{h_t}$ (see Appendix B).

the underlying latent variables equally well (Fig. 4C) with an $R^2 = 0.99$ for both, as quantified by linear probing. Since we did not observe a discernible difference between maximizing MI or not, we wondered whether the network gradients of the single variable entropy $H_R[z_t]$ and conditional entropy $H_R[z_t|z_{:t}]$, respectively, were different. We therefore computed the cosine similarity between the network gradients with respect to the kNN single-variable entropy estimator (Appendix A.5.2), and the stopgrad conditional entropy estimator (Appendix A.5.4), and found that they consistently showed good alignment (Fig. 4D).

To showcase the benefits of the Kalman-based predictor's uncertainty-aware state, which is purely based on latent-state prediction, we extended the framework to allow for input-dependent predictor parameters $\theta(x_t)$ (Appendix, Fig. A4). Doing so allows noise-aware filtering that dynamically tunes the predictor to the current input noise level. We demonstrate this benefit using an in-vivo dataset of rat hippocampal spike trains, recorded while rats ran on a linear track (Fig. 4E). Since rat traversals are periodic, Kalman-based SSL learns to arrange encoded spike trains on a circle, where trajectories approximately follow a linear system (Fig 4F). The estimated uncertainty in the Kalman filter can be directly translated into an uncertainty estimate on predictions of rat position (Fig. 4G, Appendix, Fig. A4C), which previous predictive SSL approaches do not allow.

## 7. Identifiability in predictive models

A latent representation is identifiable if it recovers the underlying latent variables up to a restricted class of transformations, rather than remaining arbitrary up to nonlinear reparameterizations. Several previous studies proved that contrastive learning can identify the true underlying variables, e.g., for pairs of inputs (Zimmermann et al., 2021) and dynamical systems (Laiz et al., 2025). In the following, we generalize these results in two respects: First, identifiability holds for any SSL variant that maximizes the LDM objective under an appropriate model, not only contrastive methods. Second, indentification requires *only* conditional distribution matching, while the marginal distribution of the latent variables may remain unconstrained.

We consider predictive models that factorize the latent distribution as $P_\theta(z) = \prod_t P_\theta(z_t|z_{:t})$, and assume that learning proceeds by matching this distribution and the empirical latent distribution $R(z)$ induced by the encoder $f$.

**Theorem 1** (Identifiability of predictive distribution matching under a Gaussian predictor). *Assume that:*

*(i) the model and true predictive distributions are Gaussian of the form:*

$$P(z_t|z_{:t}) = \mathcal{N}\big(z_t\,;\,p(z_{:t}),\Sigma\big) \quad ,$$

*with (potentially nonlinear) prediction function $p(\cdot)$*

*and non-degenerate covariance $\Sigma$;*

*(ii) the encoder is invertible on the data manifold;*

*(iii) the predictor covers the latent space.*

*Then, at the optimum of the LDM objective, the learned representation recovers the true latent variables up to an affine transformation.*

The theorem shows that predictive SSL models trained via LDM identify the actual latent state spaces, without explicit observation-level likelihoods or contrastive objectives. Identifiability arises from the local noise structure of the prediction errors: the assumed Gaussian form of the predictive residuals constrains admissible transformations of the latent space, ruling out arbitrary nonlinear reparameterizations (Fig. 5A). Importantly, the result does not require strong assumptions on the expressiveness or architecture of the predictor itself beyond coverage of the latent space.

The proof is given in Appendix A.7.1 and is closely related to previous identifiability proofs (Hyvarinen et al., 2019; Zimmermann et al., 2021; Laiz et al., 2025).

We can derive a similar identifiability result for von Mises-Fisher predictive models

**Theorem 2.** *Under assumptions (ii) and (iii) and assuming a von Mises-Fisher predictive model the LDM objective recovers the true latent space up to an affine transformation.*

See Appendix A.7.1 for the proof.

### 7.1. Empirical validation

To demonstrate identifiability in simulations we set up a nonlinear prediction task based on the simple dynamical system (Fig. 5B)

$$
\begin{cases}
\dot{x} = ak\cos(\theta)\cos(k(\theta - \theta_0)) - r\sin(\theta) + \nu_x \\
\dot{y} = ak\sin(\theta)\cos(k(\theta - \theta_0)) + r\cos(\theta) + \nu_y
\end{cases}
\tag{16}
$$

where $r = \sqrt{x^2 + y^2}$, $\theta = \arctan(y, x)$, $\theta_0$ is the angle of the system, and $k = 4$. Further, $\nu_x$ and $\nu_y$ are Gaussian distributed random variables, which are essential to satisfy Assumption *(i)* of Theorem 1—although the model is robust to limited deviations from this condition (Appendix, Fig. A6). As before, we converted positions to images of a dot, while applying an additional nonlinear swirl transformation.

To model this data, we trained MLP inference and RNN + MLP prediction networks based on $\mathcal{F}_{\mathrm{MI}}$ with a Gaussian predictor and kNN entropy estimator. This satisfies assumption (i)—note that we do not enforce assumptions (ii) and (iii), which we expected to emerge from entropy maximization. We chose a two-dimensional latent space to match the true and recovered latent dimensionality. The simulation gave two key results. First, the model linearly encodes the true

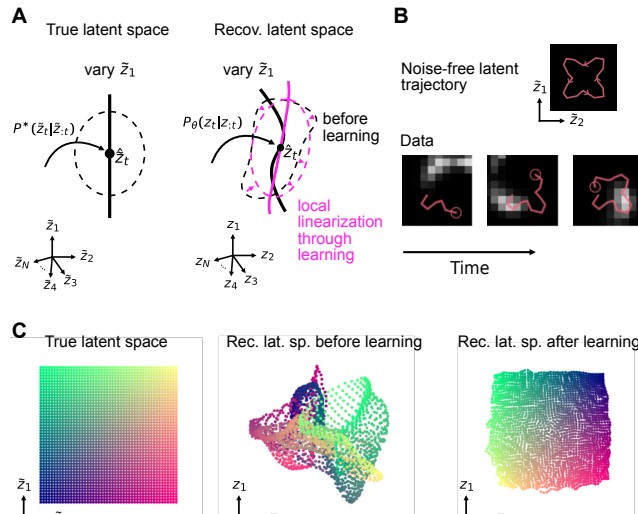

*Figure 5.* System identification through predictive LDM. **A** Forcing prediction errors into a Gaussian form leads to local linearization of the relation between true and recovered latent variables. **B** Schematic of nonlinear prediction task. Trajectory noise in the true latent space is Gaussian to enable identification. **C** Visualizations of the actual (left) and recovered latent space before (middle) and after (right) learning. Predictive LDM recovers the true space up to affine transformations.

position ($R^2 = 0.94$) and thus can recover the true latent space up to affine transformations (Fig. 5C). Second, even though the initial inference function clearly is not injective, training via entropy maximization in LDM leads to an injective encoder after learning (Appendix A.4.2).

Note that stopgrad-based models like V-JEPA make similar model assumptions as in this section, but employ a heuristic entropy estimator that assumes the conditional distribution to be Gaussian. While this was enough to identify the simpler system in Section 6, where the predictor was additionally constrained to be linear, in this nonlinear scenario, we only found approximate identification of the true system (Appendix Fig. A6).

Finally, we verified that identifiably can also be achieved with higher dimensional latent distributions, although learning becomes more challenging (Appendix, Fig. A7).

## 8. Discussion

We showed that several established approaches to SSL can be understood under the unifying principle of distribution matching in latent space. We motivated this approach with insights from linear ICA and manifold normalizing flows: For encoders that are invertible on the data manifold, maximizing the data log-likelihood is equivalent to LDM. However, instead of maximizing the log-determinant of the Jacobian, as in normalizing flows, we maximized the entropy of

the latent representations. We further showed that latent entropy maximization in LDM encourages invertibility on the data manifold. This insight explains the close connection between LDM with or without MI maximization, since in the case of an invertible encoder, the mutual information is maximized $I[f(x), f(x')] = I[x, x']$, and thus $\mathcal{F}_{\text{MI}} \propto \mathcal{F}_{\text{LDM}}$. This explains why MI-based approaches have empirically been found to enable SSL despite the invariance of MI: they systematically misestimate MI, based on constrained (e.g., Gaussian) predictors, resulting in an LDM goal.

Our analysis highlights an important conceptual point. The assumption that the encoder is approximately invertible may appear to conflict with a common intuition in SSL: that its strength lies in discarding "irrelevant" information and retaining only high-level semantic or conceptual information. The LDM framework suggests a different interpretation: SSL representations may, in fact, discard little details about the data manifold. Indeed, latent spaces in SSL models frequently span hundreds or thousands of dimensions, while the intrinsic dimensionality of image data is typically estimated to be only a few tens (Pope et al., 2021). This disparity indicates that SSL does not primarily operate through lossy compression. Rather, its advantage lies in the ability to exert fine-grained control over how the data manifold is geometrically reparameterized by specifying the model likelihood in latent, rather than data, space. By embedding appropriate inductive biases into the latent model, the resulting alignment objective can promote manifold geometries in which abstract, generalizable features are untangled and readily accessible to the predictor, while deemphasizing local, pixel-level structure that generative models prioritize to preserve.

One of the main practical challenges in the LDM approach is to design accurate, sample efficient, and easily optimizable entropy estimators. Especially the requirement of optimizability necessitates compromises when choosing between different entropy approximation approaches. For example, choosing a fixed bandwidth for KDE is not optimal for precise entropy estimation, but significantly reduces the complexity of the estimator and increases its robustness. Given that entropy estimation significantly affects representational geometry and downstream performance, designing new entropy estimators with improved performance might be one of the most pressing problems for advancing SSL. This observation aligns with recent work proposing uniformity objectives grounded in statistical test theory (Balestriero & LeCun, 2025). Viewed through the lens of LDM, such approaches can be systematically analyzed and extended in the future using tools from statistical modeling.

## Acknowledgments

We thank Steffen Schneider, Rodrigo González Laiz, Tobias Schmidt, Manu Halvagal, Atena Mohammadi, and all Zenke Lab members for their input and discussions. This project was supported by the Swiss National Science Foundation (Grant Number PCEFP3_202981) and the Novartis Research Foundation.

## Impact Statement

This paper presents work whose goal is to advance the field of machine learning. There are many potential societal consequences of our work, none of which we feel must be specifically highlighted here.

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

# Appendix Contents

# A. Mathematical appendix

## A.1. Notation

We make use of the following mathematical notation

- $\langle f(x) \rangle_{P(x)} = \int f(x) P(x) \, dx$ denotes the average of $f(x)$ w.r.t. $P(x)$.

- $\delta(x)$ is the Dirac delta function.

- $H_P[x] = -\int P(x) \log P(x) \, dx$ is the entropy of $P(x)$.

- $H_P[x|y] = -\int P(x,y) \log P(x|y) \, dx$ is the conditional entropy of $P(x|y)$.

- $I_P[x,y] = H_P[x] + H_P[y] - H_P[x,y]$ is the mutual information between $x$ and $y$.

- $D_{KL}[P(x) \parallel Q(x)] = \int P(x) \log(P(x)/Q(x)) \, dx$ is the Kullback-Leibler divergence of $P(x)$ and $Q(x)$.

- $|M|$ denotes the determinant of the matrix $M$.

- $J_f(x) = \left[ \frac{\partial f(x)}{\partial x_1} \ldots \frac{\partial f(x)}{\partial x_n} \right]$ is the Jacobian matrix of $f$ evaluated at $x$.

- $\mathrm{SG}[\cdot]$ is the stop-gradient operator, which prevents optimization gradients from flowing through this node

## A.2. Related frameworks

### A.2.1. REVERSE KL DIVERGENCE

A formalism closely related to ours was proposed by Zimmermann et al. (2021). In their theory, the latent distribution is defined through the conditional $R(z'|z) = \frac{1}{Z} \exp(-E(f(x), f(x')))$, where $E(\cdot)$ is an energy function and $f$ encodes $x$ to $z$. The goal function is then (implicitly) defined as

$$\mathcal{F}_{\mathrm{MDL}} = -D_{KL}[P_\theta(z'|z) \parallel R(z'|z)] \propto \langle \log R(z'|z) \rangle_{P_\theta(z'|z)} , \qquad (17)$$

where the proportionality holds for fixed $\theta$. This is also called the *reverse KL divergence* (Papamakarios et al., 2021). While here it is not necessary to compute an entropy term, the difficulty with this approach is that, instead of averaging over the data distribution, it requires to average over the latent distribution $P_\theta(z, z')$. This average can only be taken in specific circumstances, e.g., for uniform distributions on the sphere, which hinders a more general applicability of the framework, and renders relating it to the diversity of SSL approaches difficult.

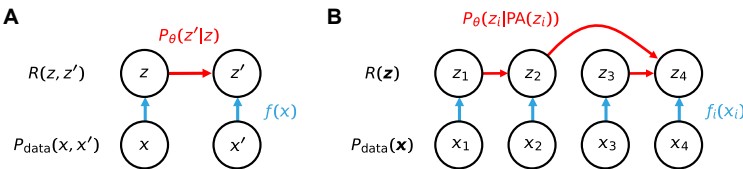

*Figure A1.* Examples of LDM models. **A** In a simple model with two inputs the goal is to predict one latent variable from the other. **B** In general, any loop-free graphical model can form the predictor on any number of inputs, possibly with distinct encoders $f_i(x_i)$.

### A.2.2. I-Con

Another similar loss has been proposed by Alshammari et al. (2025). I-Con defines two conditional neighborhood distributions over data points $i, j \in \mathcal{X}$: a supervisory distribution $p_\theta(j|i)$ encoding assumed relationships between samples (e.g., augmentation pairs, class labels, $k$-nearest neighbors), and a learned distribution $q_\phi(j|i)$ derived from the representations. The I-Con loss then takes the form

$$\mathcal{L}_{\text{I-Con}} = \int_{i \in \mathcal{X}} D_{\text{KL}} \left( p_\theta(\cdot|i) \| q_\phi(\cdot|i) \right). \tag{18}$$

A key structural difference to LDM is that I-Con treats the supervisory signal $p_\theta(j|i)$ as given (or separately constructed), and optimizes the learned representation to match it. LDM instead specifies a generative latent model $P_\theta(\mathbf{z}'|\mathbf{z})P_\theta(\mathbf{z})$ and jointly optimizes both the encoder and the model parameters $\theta$ to minimize the divergence in latent space. An unresolved question is wether I-Con can also establish identifiability guarantees, similar to the forward or reverse KL divergence approaches.

### A.2.3. Variational bound on mutual information

Here we clarify the relation of the mutual information bound used in the main paper, and the bound most previous work employs (Barber & Agakov, 2004; Poole et al., 2019; Wang & Isola, 2020; Shwartz-Ziv et al., 2023; Gálvez et al., 2023). This second bound takes the form

$$I_R[z, z'] \geq \hat{I}_R[z, z'] = \langle \log P_\theta(z|z') \rangle_{R(z,z')} + H_R[z], \tag{19}$$

which seems to differ from $\mathcal{F}_{\text{MI}}$. However, these two MI bounds are easy to relate. To show this, we assume that the latent model employs an empirical prior, and thus takes the form $P_\theta(z, z') = P_\theta(z|z')R(z')$. Under this model $\mathcal{F}_{\text{MI}}$ simplifies to

$$\begin{aligned}
\mathcal{F}_{\text{MI}} &= -D_{KL}[R(z, z') \| P_\theta(z, z')] + I_R[z, z'] \\
&= \langle \log P_\theta(z, z') \rangle_{R(z,z')} + H_R[z] + H_R[z'] \\
&= \langle \log P_\theta(z|z') + \log R(z') \rangle_{R(z,z')} + H_R[z] + H_R[z'] \\
&= \langle \log P_\theta(z|z') \rangle_{R(z,z')} + H_R[z].
\end{aligned} \tag{20}$$

### A.3. Latent distribution matching framework overview

In its most general form the goal function can be written as

$$\begin{aligned}
\mathcal{F}_{\text{LDM}} &= -D_{KL}[R(\mathbf{z}) \| P_\theta(\mathbf{z})] \\
&= \underbrace{\left\langle \sum_i \log P_\theta(z_i|\text{PA}(z_i)) \right\rangle_{R(\mathbf{z})}}_{\text{Likelihood}} + \underbrace{H_R[\mathbf{z}]}_{\text{Entropy}}
\end{aligned} \tag{21}$$

where $\mathbf{z} = \{z_i : i \in [1 \ldots N]\}$ and $\text{PA}(z_i)$ are the parents of $z_i$ in the probabilistic model. The parents of $z_i$ can also include dependencies on additional conditioning variables that are independent of other observations, which we did not consider

in this paper. These can be, for example, control signals, time indices, etc. While the joint entropy $H_R[\boldsymbol{z}]$ can be difficult to maximize, given the arguments in this paper (Section A.4.3), it might be replaced by the sum of individual entropies $\sum_i H_{R_i}[z_i]$ with indistinguishable results.

The empirical latent distribution induced by the data distribution and the encoder is defined by

$$
R(\boldsymbol{z}) = \left\langle \prod_i R_i(z_i | x_i) \right\rangle_{P_{\text{data}}(\boldsymbol{x})} \tag{22}
$$

This formulation also subsumes models with multimodal inputs and multiple encoders, or temporal models (Figure A1). In the following we only employ deterministic encoders $R_i(z_i | x_i) = \delta(z_i - f_i(x_i))$, as motivated by the discussion in Section A.4.1, while probabilistic encoders are discussed in Section A.8.

## A.4. Motivation of latent distribution matching framework

### A.4.1. RELATION BETWEEN MAXIMUM LIKELIHOOD LEARNING AND DISTRIBUTION MATCHING

We can motivate the LDM framework through the maximum likelihood objective, mirroring the approach employed in ICA (Hyvärinen & Oja, 1999) or normalizing flow networks (Papamakarios et al., 2021). Here, models often compute the data-likelihood in latent, instead of data space, by applying the change of variables formula. Specifically, if the transformation $g$ from latent variables $z$ to data $x$ is invertible

$$
\begin{aligned}
\mathcal{L}(\theta, g) &= \langle \log P_\theta(x) \rangle_{P_{\text{data}}(x)} \\
&= \left\langle \log P_\theta(g^{-1}(x)) + \log |J_{g^{-1}}(x)| \right\rangle_{P_{\text{data}}(x)} ,
\end{aligned} \tag{23}
$$

where $P_\theta$ is the model distribution with parameters $\theta$ and $J_{g^{-1}}(x)$ is the Jacobian of the inverse transformation (see also Papamakarios et al., 2021). Here and in the following we assume that $g$ (and $f$) is a smooth and continuous function. The change of variables as above requires data and latent space to be of the same dimensionality and thus cannot be applied to functions that change the number of dimensions, as employed here. However, for the following argument it is sufficient if the encoder is invertible *on the data manifold* $\mathcal{M}_{\text{data}}$, which is also possible if the data manifold is embedded into a higher dimensional space. This can be shown via a generalized change of variables formula that has been previously used, e.g., in manifold normalizing flows (Brehmer & Cranmer, 2020), and takes the form

$$
\begin{aligned}
\mathcal{L}(\theta, g) &= \langle \log P_\theta(x) \rangle_{P_{\text{data}}(x)} \\
&= \left\langle \log P_\theta(g^{-1}(x)) + \frac{1}{2} \log |J_{g^{-1}}(x) J_{g^{-1}}(x)^T| \right\rangle_{P_{\text{data}}(x)} .
\end{aligned} \tag{24}
$$

We can relate the maximum likelihood objective to a LDM goal by noticing that for any transformation $f$ that is invertible on the data manifold, the differential entropy changes as

$$
H_P[x] = H_P[f(x)] - \left\langle \frac{1}{2} \log |J_f(x) J_f(x)^T| \right\rangle_{P(x)} , \tag{25}
$$

which is also a consequence of the change of variables formula for manifolds. Thus, if we define the encoder $f(x) = g^{-1}(x)$ we can rewrite the log-likelihood in an exclusively latent-space form, by adding the constant data entropy (Papamakarios et al., 2021; Cardoso, 2002)

$$
\begin{aligned}
\mathcal{L}(\theta, f) &\propto \left\langle \log P_\theta(f(x)) + \frac{1}{2} \log |J_f(x) J_f(x)^T| \right\rangle_{P_{\text{data}}(x)} + H_{P_{\text{data}}}[x] \\
&= \langle \log P_\theta(f(x)) \rangle_{P_{\text{data}}(x)} + H_{P_{\text{data}}}[f(x)] \\
&= -D_{KL}[P_{\text{data}}(f(x)) \| P_\theta(f(x))] .
\end{aligned} \tag{26}
$$

In summary, for an invertible data generation process, maximum likelihood learning is equivalent to LDM in latent space. This also means that, in principle, we can estimate the log-likelihood of a datapoint based on the change of variables formula

above; we do not explore this here, but see Balestriero et al. (2025). The other observation we want to highlight is that, while manifold normalizing flows is concerned with maximizing or regularizing the Jacobian term $\log |J_f(x)J_f(x)^T|$, it is equivalent to maximize the latent entropy instead. Notably, while commonly manifold normalizing flow networks are constrained to be injective on the data manifold, we do not constrain the networks, but instead fully rely on the regularization through entropy maximization (Section A.4.2). This is closely related to approaches in manifold normalizing flow that aim to drop architectural constraints to improve expressibility (Sorrenson et al., 2023).

### A.4.2. APPROXIMATE INVERTIBILITY THROUGH ENTROPY MAXIMIZATION

In the proposed goal function the encoder is encouraged to be both locally and globally invertible on the data manifold through the entropy term. To see how it encourages local invertibility, we assume the data lies on a manifold $\mathcal{M}_{\text{data}}$ of intrinsic dimension matching the latent space. Here we define the manifold as the support of the data distribution $\mathcal{M}_{\text{data}} = \text{supp}(P_{\text{data}})$. For the encoder to be a local diffeomorphism (invertible on the manifold), the Jacobian restricted to the tangent space must have full rank (Krantz & Parks, 2008, p. 125). This is satisfied if for all $x \in \mathcal{M}_{\text{data}}$ (Inverse function theorem):

$$
\sqrt{|J_f(x)J_f(x)^T|} > 0
$$
$$
\Leftrightarrow \quad \frac{1}{2} \log |J_f(x)J_f(x)^T| > -\infty \tag{27}
$$

On the other hand, through a generalized change of variables formula for manifolds, the latent entropy has the upper bound (Geiger & Kubin, 2012):

$$
H_P[f(x)] \leq H_P[x] + \left\langle \frac{1}{2} \log |J_f(x)J_f(x)^T| \right\rangle_{P(x)}, \tag{28}
$$

which is an equality for injective (non-folding) transformations. Consequently, if the encoder is locally invertible over the data distribution upon initialization, latent entropy maximization encourages it to remain locally invertible during training by penalizing singular Jacobians, i.e., for which the log-determinant would approach $-\infty$ (see also Fig. A2A).

Global invertibility on the data manifold of $f$ requires additional assumptions next to non-singular Jacobians, for example that $f$ is a proper map as assumed in *Hadamard's global inverse function theorem* (Krantz & Parks, 2002, p. 127). However, entropy maximization also provides an inductive bias towards global invertibility. Consider a mapping that is locally invertible but globally non-injective (e.g., a 'folded' or 'wrapped' manifold). Such a mapping inevitably superimposes probability mass from distinct data regions onto the same latent locations. If the amount the distribution can be 'stretched' by $f$ (the log-determinant of the Jacobian) is limited, which typically is a result of simultaneously maximizing the model log-likelihood, this creates regions of higher probability density. Since differential entropy is maximized by spreading probability mass as uniformly as possible over the available volume, any such superposition (folding) results in a suboptimal entropy compared to an unfolded, globally injective representation.

To be more precise, how much information is lost by such a folding of the manifold is quantified by the missing term in the previous inequality (Equation 28), which is also termed the *folding entropy* $H_P[x|f(x)]$. This leads to the expression of the latent entropy, which—for finite Jacobian—is maximized for non folding manifolds with $H_P[x|f(x)] = 0$ (Geiger & Kubin, 2012; Ruelle, 1996)

$$
H_P[f(x)] = H_P[x] + \left\langle \frac{1}{2} \log |J_f(x)J_f(x)^T| \right\rangle_{P(x)} - H_P[x|f(x)] \tag{29}
$$

This equation holds for piecewise invertible $f$, i.e., most non-degenerate neural networks—see also the derivation in Section A.7.2. Thus, the objective function naturally penalizes topological defects and encourages the encoder to effectively 'unfold' the data manifold into the latent space. Note that LDM might still lead to non-injective encoders under a misspecification of the model, e.g., if the prior $P_\theta(z)$ is chosen too narrowly, or the conditional $P_\theta(z'|z)$ is too loose and permits extreme stretching of the manifold—or, of course, if entropy estimation is inaccurate.

Previously, invertibility of the encoder on the data manifold has been proven to result from contrastive learning under different assumptions by Zimmermann et al. (2021); Schneider et al. (2023); Reizinger et al. (2024).

### A.4.3. MAXIMIZATION OF MUTUAL INFORMATION THROUGH ENTROPY MAXIMIZATION

Given the argument in A.4.2, we can link MI maximization between related inputs to latent entropy maximization. First, we note that through the data processing inequality we know that for any encoder $f$ MI can only decrease

$$I[x, x'] \geq I[f(x), f(x')] . \tag{30}$$

As discussed in the introduction, a fundamental fact about MI is that if the encoder $f$ is invertible on the data manifold, then MI is preserved

$$I[x, x'] = I[f(x), f(x')] . \tag{31}$$

Clearly, this is the maximum MI achievable. As discussed in A.4.2, invertibility can be encouraged simply by maximization of the latent entropy (while restricting the stretching of the latent manifold), which thus also implies approximate maximization of MI.

## A.5. Entropy estimators

Before discussing the derivations of specific goal functions, we briefly introduce the entropy estimators used to approximate the uniformity terms.

### A.5.1. KERNEL DENSITY ESTIMATION (KDE)

KDE first approximates the continuous probability density function (PDF) $R(z)$ through probability kernels $\kappa$ around samples $z_j \sim R(z)$, and based on this approximates the entropy (Ahmad & Lin, 1976). The kernel approximation of the pdf is given by $\hat{R}(z) = \frac{1}{nh} \sum_j \kappa \left( \frac{d(x, z_j)}{h} \right)$, where $d(\cdot, \cdot)$ is some distance measure, $n$ the number of samples, and $h$ is the bandwidth of the kernel. The entropy estimate is then given by

$$\hat{H}_{\text{KDE}}[z] = -\frac{1}{n} \sum_i \log \hat{R}(z_i) \propto -\frac{1}{n} \sum_i \log \sum_j \kappa \left( \frac{d(z_i, z_j)}{h} \right) . \tag{32}$$

Note that we immediately can identify the comparison of $z_i$ and $z_j$ with the contrastive principle of SSL.

### A.5.2. K-NEAREST NEIGHBOR (KNN) ESTIMATION

Another robust estimator for the entropy of high-dimensional distributions is the Kozachenko and Leonenko nearest neighbor estimator (Kozachenko, 1987), which later has been generalized to kNN estimators (Lombardi & Pant, 2016). The estimator constructs a ball around every sample $z_i$, with radius given by the $p$-norm distance $\epsilon_i = |z_i - \text{kNN}(z_i)|_p$ between $z_i$ and its kNN. Similar to the KDE, the sum of balls then approximate the PDF under the assumption of uniform distribution within the balls. This gives rise to the entropy estimator

$$\hat{H}_{\text{kNN}}[z] \propto \frac{D}{n} \sum_i \log \epsilon_i . \tag{33}$$

One advantage of this estimator over KDE is that its parameters $p$ and $k$ are more straightforward to choose than the bandwidth $h$.

### A.5.3. PARAMETRIC ESTIMATION

Entropy can also be estimated via a parametric approach, by assuming that the data is distributed according to a distribution with a closed form entropy expression $\hat{R}_\phi(z)$, and estimating the parameters $\phi$ of this distribution from the data $z_j \sim R(z)$. In practice, to the best of our knowledge, the only non-trivial distribution that allows to feasibly compute the entropy in high dimensions is the multidimensional normal and derived distributions (e.g., log-normal). For a $D$-dimensional normal $\hat{R}_\phi(z) = \mathcal{N}(z; \mu, \Sigma)$ we can estimate $\mu = \frac{1}{N} \sum_j z_j$ and $\Sigma = \frac{1}{N-1} \sum_j (z_j - \mu)(z_j - \mu)^T$, and the entropy is given by the log-determinant of the covariance matrix

$$\hat{H}_{\text{LogDet}}[z] = \frac{D}{2}(1 + \log(2\pi)) + \frac{1}{2} \log |\Sigma| . \tag{34}$$

In practice, for high dimensional distributions, the determinant is numerically complex to compute, and additional approximations are needed (Zhouyin & Liu, 2025). To provide additional intuition, a very simple approximation can be obtained by assuming that off-diagonal terms are small, $\Sigma = D + A$, with $D$ diagonal and $A$ small. Performing a low-order Taylor expansion yields $\log |\Sigma| = \sum_i \log \Sigma_{ii} - \frac{1}{2} \sum_{j \neq i} \frac{\Sigma_{ij}\Sigma_{ji}}{\Sigma_{ii}\Sigma_{jj}} + \mathcal{O}(A^3)$. From this it becomes clear that entropy is maximized for maximum variance and minimum covariance terms.

### A.5.4. CONDITIONAL ENTROPY ESTIMATION WITH PREDICTOR

The conditional entropy of $R(z'|z)$ is given by $H_R[z'|z] = -\langle \log R(z'|z) \rangle_{R(z',z)}$. Here we will evaluate the average $\langle \cdot \rangle_{R(z',z)}$ via samples, and the main question regards how the conditional distribution is approximated. The most straightforward approach is to learn a predictor $\hat{R}_\theta(z'|z)$ by minimizing the cross entropy $-\langle \log \hat{R}_\theta(z'|z) \rangle_{R(z',z)}$ with respect to the parameters $\theta$. The entropy can then be approximated and maximized with the plug-in estimator

$$\hat{H}_{\text{pred}}[z'|z] = -\frac{1}{N} \sum_{(z',z)} \log \hat{R}_\theta(z'|z) . \tag{35}$$

Intriguingly, the cross entropy to train the predictor is exactly equal to the entropy estimator of the conditional entropy. This means that when we train a predictor $\hat{R}_\theta(z'|z)$ via cross-entropy, we are simultaneously obtaining an entropy estimate for free, without needing a separate estimator. We will discuss how this observation is leveraged implicitly by SSL models with predictor and stopgrad in Section A.6.2.

### A.6. Goal function derivations

#### A.6.1. MODELS WITH PAIRS OF INPUTS

We will first treat models with pairs of inputs, and later discuss temporal models. Specifically, the goal is to derive the likelihood and entropy terms of the goal functions, which are special cases of the general goal function (Eq 21)

$$\mathcal{F}_{\text{LDM}} = \langle \log P_\theta(z, z') \rangle_{R(z,z')} + H_R[z, z'] \tag{36}$$

$$\mathcal{F}_{\text{MI}} = \langle \log P_\theta(z, z') \rangle_{R(z,z')} + 2H_R[z] \tag{37}$$

Given the definition of the induced latent distribution (Eq 22) and deterministic encoders, the average likelihood becomes

$$\begin{aligned}
\langle \log P_\theta(z, z') \rangle_{R(z,z')} &= \int (\log P_\theta(z, z') \langle R(z|x)R(z'|x') \rangle_{P_{\text{data}}(x,x')} \, dz \, dz' \\
&= \left\langle \int (\log P_\theta(z, z'))R(z|x)R(z'|x') \, dz \, dz' \right\rangle_{P_{\text{data}}(x,x')} \\
&= \langle \log P_\theta(f(x), f(x')) \rangle_{P_{\text{data}}(x,x')} \\
&= \langle \log P_\theta(f(x')|f(x)) + \log P_\theta(f(x)) \rangle_{P_{\text{data}}(x,x')} ,
\end{aligned} \tag{38}$$

where in the third step we used $R(z|x) = \delta(z - f(x))$. In the last step we used $P_\theta(z, z') = P_\theta(z'|z)P_\theta(z)$.

The following derivations will first specify assumptions for $P_\theta(z'|z)$ and $P_\theta(z)$, and then approximations for the entropy, based on the estimators outlined in detail in Section A.5. See also the derivations in Wang & Isola (2020); Shwartz-Ziv et al. (2023); Gálvez et al. (2023), which employed the MI estimator derived from $\mathcal{F}_{\text{MI}}$ in A.2.3.

**VICReg** We make the choices

$$\begin{aligned}
P_\theta(z) &= Flat, \\
P_\theta(z'|z) &= \mathcal{N}(z'; \mu = z, \Sigma = \sigma^2 I) .
\end{aligned} \tag{39}$$

The likelihood term simplifies to

$$\begin{aligned}
&\langle \log P_\theta(f(x')|f(x)) + \log P_\theta(f(x)) \rangle_{P_{\text{data}}(x,x')} \\
&= \langle \log \mathcal{N}(f(x'); \mu = f(x), \Sigma = \sigma^2 I) \rangle_{P_{\text{data}}(x,x')} \\
&\propto -\left\langle \frac{1}{2\sigma^2} \| f(x) - f(x') \|^2 \right\rangle_{P_{\text{data}}(x,x')} .
\end{aligned} \tag{40}$$

To estimate the entropy, we approximate the latent distribution with a Normal distribution $R(z, z') \approx \mathcal{N}([z, z']; \mu_{z,z'}, \Sigma_{z,z'})$ or $R(z) \approx \mathcal{N}(z; \mu_z, \Sigma_z)$ with $\mu$ the empirical mean and $\Sigma$ the empirical covariance matrix. This leads to the closed form entropy approximation

$$
H_R[z, z'] \propto \frac{1}{2} \log |\Sigma_{z,z'}|
$$

$$
H_R[z] \propto \frac{1}{2} \log |\Sigma_z| \ .
$$

$$(41)$$

While there is no closed form for the determinant of the covariance matrix that can be optimized easily, it is clear that a maximum entropy Gaussian is achieved with maximal variance and minimal off-diagonal covariance terms. One potential approach to show this is to assume that the covariance is close to diagonal ($D$ diagonal and $A$ small) and expand the log-determinant

$$
\log |\Sigma| = \log |D + A| \approx \sum_i \log \Sigma_{ii} - \sum_{i \neq j} C_{ij} \ ,
$$

$$(42)$$

where the first term promotes large variance and the second small covariance $C_{ij} = \Sigma_{ij}^2 / (\Sigma_{ii} \Sigma_{jj})$ terms. Overall, VICReg can be understood to implement an approximation to the goal of minimizing

$$
\mathcal{F}_{\mathrm{MI}} = -\frac{1}{2\sigma^2} \left\langle \| f(x) - f(x') \|^2 \right\rangle_{P_{\mathrm{data}}(x,x')} + \log |\Sigma_z| \ .
$$

$$(43)$$

See also Shwartz-Ziv et al. (2023), which finds a similar goal function based on principles of MI maximization.

**SimCLR**   We make the choices, with latents $z$ lying on the unit sphere

$$
P_\theta(z) = \frac{1}{Z},
$$

$$
P_\theta(z'|z) = \frac{1}{Z} \exp(\beta z^T z') \ .
$$

$$(44)$$

The likelihood term then becomes

$$
\left\langle \iint_S (\log P_\theta(z, z')) R(z|x) R(z'|x') \, dz \, dz' \right\rangle_{P_{\mathrm{data}}(x,x')}
$$

$$
\propto \left\langle \int_S \beta z^T z' \delta(z - f(x)) \delta(z' - f(x')) \, dz \, dz' \right\rangle_{P_{\mathrm{data}}(x,x')}
$$

$$
= \left\langle \beta f(x)^T f(x') \right\rangle_{P_{\mathrm{data}}(x,x')} \ .
$$

$$(45)$$

The Entropy term becomes, using the KDE approach with von Mises-Fisher kernels with bandwidth $\gamma$

$$
\hat{H}_{\mathrm{KDE}}[z] \propto -\frac{1}{N} \sum_i \log \left[ \sum_j \kappa(f(x_i), f(x_j), \gamma) \right]
$$

$$
= -\frac{1}{N} \sum_i \log \left[ \sum_j 1/Z \exp(\gamma^{-1} f(x_i)^T f(x_j)) \right]
$$

$$
\propto -\frac{1}{N} \sum_i \log \left[ \sum_j \exp(\gamma^{-1} f(x_i)^T f(x_j)) \right] \ .
$$

$$(46)$$

If the entropy is approximated via KDE with bandwidth $\gamma = 1/\beta$ then based on $\mathcal{F}_{\mathrm{LDM}}$ we find the goal function

$$
\mathcal{F} = \left\langle \beta f(x)^T f(x') \right\rangle_{P_{\mathrm{data}}(x,x')} - \left\langle \log \langle \exp\{\beta(f(x)^T f(x^-) + f(x')^T f(x'^-))\} \rangle_{P_{\mathrm{data}}(x^-, x'^-)} \right\rangle_{P_{\mathrm{data}}(x,x')} \ .
$$

$$(47)$$

Similar to VICReg, the original goal function can be recovered from $\mathcal{F}_{\mathrm{MI}}$ instead as

$$
\mathcal{F} = \left\langle \beta f(x)^T f(x') \right\rangle_{P_{\mathrm{data}}(x,x')} - 2 \left\langle \log \langle \exp\{\beta f(x)^T f(x^-)\} \rangle_{P_{\mathrm{data}}(x^-)} \right\rangle_{P_{\mathrm{data}}(x)} \ .
$$

$$(48)$$

Technically speaking, the SimCLR goal differs from this goal function in that in SimCLR the negative samples $x^-$ include $x'$, the positive sample, while in LDM all samples in the entropy are i.i.d.—in practice, however, we did not observe a significant difference between the two.

**SSL on the simplex** Next to latent variables on the plane and sphere, we can also consider variables on the simplex $\Delta^{D-1} = \{z \in \mathbb{R} : \sum_i z_i = 1\}$, which can be obtained with a softmax output layer. Often, points on the simplex are understood as probability distributions over categorical variables, but we will treat them here simply as points, allowing us to understand encoders as deterministic functions, as before. This leads to loss functions related to those used in the DINO family (Caron et al., 2021).

We make the choices, with latents $z$ lying on the simplex

$$
P_\theta(z) = \frac{1}{Z},
$$
$$
P_\theta(z'|z) = \text{Dir}(z', \tau z + 1) = \frac{1}{B(\tau z + 1)} \prod_k z_k'^{\tau z_k} .
$$
(49)

where here and in the following $k$ indexes the *components* of z, $\text{Dir}(p, \alpha) = \frac{1}{B(\alpha)} \prod_k z_k^{\alpha_k - 1}$ is the Dirichlet distribution, and $B(\alpha) = \frac{\prod_k \Gamma(\alpha_k)}{\Gamma(\sum_k \alpha_k)}$. In this model, $\tau$ can be understood as a precision parameter, similar to precision in, e.g., a Gaussian distribution. Note that the mean of the conditional distribution is $\frac{\alpha}{\sum_k \alpha_k} = \frac{\tau z + 1}{\tau + K}$, but the mode is $\frac{\alpha - 1}{\sum_k \alpha_k - K} = z$, justifying this choice of model.

The likelihood term for this model (dropping constant terms) is

$$
\langle \log P_\theta(z, z') \rangle_{R(z,z')} \propto \left\langle \tau z^T \log(z') - \sum_k \log \Gamma(\tau z_k + 1) \right\rangle_{R(z,z')} ,
$$
(50)

With Dirichlet-based KDE entropy, we can derive

$$
\hat{H}_{\text{KDE}}[z] \propto -\frac{1}{N} \sum_i \log \left[ \sum_j \text{Dir}(z_i, \tau z_j) \right]
$$
$$
\propto -\frac{1}{N} \sum_i \log \left[ \sum_j \exp \left( \tau z_j^T \log(z_i) - \sum_k \log \Gamma(\tau z_{jk} + 1) \right) \right] .
$$
(51)

Taken together we arrive at the DINO alignment term $\tau \sum_k z_k \log(z_k')$, and an effective collapse prevention term that depends on contrastive samples and the log Gamma term from the normalization. The DINO models, in contrast, rely on more heuristic collapse prevention methods, that might not directly be related to any entropy estimator.

Empirically, we find that the other entropy estimators (kNN, LogDet) are not suited for variables on the simplex and do not converge. Results for the contrastive approach are summarized in Table A1.

### A.6.2. PREDICTIVE STOPGRAD MODELS

We can also understand predictive models with stopgrad (Grill et al., 2020; Chen et al., 2020; Bardes et al., 2024; Assran et al., 2025; Mohammadi et al., 2025) as approximately maximizing entropy. These models have the general goal function

$$
\mathcal{F} = \sum_t \langle \log P_\theta(SG[z_t]|z_{:t}) \rangle_{R(z_t, z_{:t})} ,
$$
(52)

where commonly, but not necessarily, $P_\theta$ is a Normal distribution with either fixed or variable variance. To recall, in the proposed framework of LDM, the goal for a temporal model is the same as before, with the general goal function

$$
-D_{KL}[R(\boldsymbol{z}) \| P_\theta(\boldsymbol{z})] = \sum_t \langle \log P_\theta(z_t|z_{:t}) \rangle_{R(z_t, z_{:t})} + H_R[z_t|z_{:t}] .
$$
(53)

To show the relation to predictive stopgrad losses, we start with the basic observation that

$$
\langle \log P_\theta(SG[z_t]|z_{:t}) \rangle_{R(z_t, z_{:t})} \stackrel{\partial}{=} \langle \log P_\theta(z_t|z_{:t}) \rangle_{R(z_t, z_{:t})} - \langle \log P_{SG[\theta]}(z_t|SG[z_{:t}]) \rangle_{R(z_t, z_{:t})} .
$$
(54)

where $\stackrel{\partial}{=}$ denotes equal in derivative. Thus, both goal functions match in derivative if the second term approximates the (derivative of the) conditional entropy of $R(z_t|z_{:t})$. First, we can understand $P_{SG[\theta]}(\cdot|\cdot) = \hat{R}(\cdot|\cdot)$ as a *fixed* estimator of the conditional distribution $R(z_t|z_{:t})$, since it is only trained via the first term which attains a minimum if the conditional distributions match $P_{SG[\theta]}(\cdot|\cdot) = R(\cdot|\cdot)$. Note, that this requires to train $P_\theta$ on a faster timescale than the encoders. Some stopgrad based approaches ensure this heuristically by employing exponentially moving average encoder targets in the goal function (Grill et al., 2020; Assran et al., 2023). Note also that there is no stopgrad on the average, since $R$ is not parametrized directly, but is defined through sampling $x$ and mapping from $x$ to $z$. We thus see that the second term is a sampling-based estimator of the conditional entropy (see also Section A.5.4)

$$\langle -\log P_{SG[\theta]}(z_t|SG[z_{:t}])\rangle_{R(z_t, z_{:t})} = \left\langle -\log \hat{R}(z_t|SG[z_{:t}])\right\rangle_{R(z_t, z_{:t})} \approx H_R[z_t|SG[z_{:t}]] . \tag{55}$$

The stopgrad approach can therefore be seen to approximately maximize conditional entropy via $z_t$. We can gain more insight by rewriting $H_R[z_t|SG[z_{:t}]] = H_R[z_t] - I_R[z_t, SG[z_{:t}]]$. This shows that, in fact, the single timestep entropy is maximized fully, while the term that is only partially minimized is the (as we have argued, typically inconsequential) MI term. This suggests that in principle it would also be feasible to train a separate (ideally, less constrained) predictor $\hat{R}_\phi(z_t|z_{:t})$ with parameters $\phi$ on a faster timescale and use this *fixed* predictor to estimate the conditional entropy. We leave this to be explored by future research.

**BYOL/SimSiam** We first apply this idea to models in the nontemporal setting. To derive the goal function of BYOL (Grill et al., 2020) and SimSiam (Chen et al., 2020) we make the choices

$$\begin{aligned} P_\theta(z) &= R(z), \\ P_\theta(z'|z) &= \frac{1}{Z} \exp(\tau p(z)^T z') , \end{aligned} \tag{56}$$

where $p(\cdot)$ is the predictor, and latent variables are normalized to lie on the sphere. Here an empirical prior on latent variables is assumed, similar to the model in Aitchison & Ganev (2024). Future work might explore models with a proper prior to reduce the need for additional model regularization (Grill et al., 2020). With the empirical prior $\mathcal{F}_{\text{LDM}}$ simplifies to

$$\begin{aligned} \mathcal{F}_{\text{LDM}} &= -D_{KL}[R(z'|z)R(z) \parallel P_\theta(z'|z)R(z)] \\ &= -\langle D_{KL}[R(z'|z) \parallel P_\theta(z'|z)]\rangle_{R(z)} \\ &= \langle \log P_\theta(z'|z)\rangle_{R(z, z')} + H_R[z'|z] . \end{aligned} \tag{57}$$

Here we employ the same trick as in A.6.2 and replace the conditional entropy with the stopgrad entropy estimator. This recovers the original goal function

$$\begin{aligned} \mathcal{F}_{\text{LDM}} &\approx \langle \log P_\theta(z'|z)\rangle_{R(z, z')} - \langle \log P_{SG[\theta]}(z'|SG[z])\rangle_{R(z, z')} \\ &\stackrel{\partial}{=} \langle \log P_\theta(SG[z']|z)\rangle_{R(z, z')} \\ &\propto \left\langle \tau p(z)^T SG[z']\right\rangle_{R(z, z')} . \end{aligned} \tag{58}$$

**Temporal Gaussian models** The same argument can be made for temporal models, or models where the predictor is conditioned on multiple variables $z_{:t}$ (Bardes et al., 2024; Assran et al., 2025; Mohammadi et al., 2025). To derive these methods we make the choices

$$\begin{aligned} P_\theta(z_0) &= R(z_0), \\ P_\theta(z_t|z_{:t}) &= \mathcal{N}(z_t; \mu = p(z_{:t}), \Sigma = \sigma^2 I) , \end{aligned} \tag{59}$$

where $p(\cdot)$ now is a temporal/multi-input predictor, specified, e.g., by an RNN. With the same argument as above we find the goal function

$$
\begin{aligned}
\mathcal{F}_{\text{LDM}} &= \sum_t \langle \log P_\theta(z_t|z_{:t}) \rangle_{R(z_t, z_{:t})} + H_R[z_t|z_{:t}] \\
&\approx \left\langle \sum_t \log P_\theta(z_t|z_{:t}) \rangle_{R(z, z')} - \log P_{SG[\theta]}(z_t|SG[z_{:t}]) \right\rangle_{R(z)} \\
&\overset{\partial}{=} \left\langle \sum_t \log P_\theta(SG[z_t]|z_{:t}) \right\rangle_{R(z)} \\
&\propto \left\langle -\sum_t \frac{1}{2\sigma^2} \|SG[z_t] - p(z_{:t})\|^2 \right\rangle_{R(z)} .
\end{aligned}
\tag{60}
$$

## A.7. Proofs

### A.7.1. IDENTIFIABILITY OF PREDICTIVE MODEL

Our goal is to show that under a predictive Gaussian model LDM recovers the original latent variables up to trivial transformations.

**Definition 1** (Data generation process). We start by defining the true latent variable distribution

$$
\begin{aligned}
P(\tilde{z}) &= \prod_t P(\tilde{z}_t|\tilde{z}_{:t}) \\
P(\tilde{z}_t|\tilde{z}_{:t}) &= \mathcal{N}(\tilde{z}_t; p(\tilde{z}_{:t}), \Sigma) .
\end{aligned}
\tag{61}
$$

Note that here $p$ and $\Sigma$ can be time dependent, which we do not write down explicitly for compactness. We assume an injective data generation process $g$, which together with the latent distribution defines the data distribution.

**Definition 2.** The model is defined by the encoder $f$ and the data distribution. This leads to the transformation between true and recovered latent variables $h = f \circ g$. Further, by assumption of a sufficiently flexible encoder $f$, after LDM to the form of (61) the latent distribution has the form

$$
\begin{aligned}
R(z) &= \prod_t R(z_t|z_{:t}) \\
R(z_t|z_{:t}) &= \mathcal{N}(z_t; p_R(z_{:t}), \Sigma_R) .
\end{aligned}
\tag{62}
$$

Note that, technically speaking, $p_R$ is the predictor that is *induced* by $f$. This is different from the predictor that would be learned in the target distribution of the model $P_\theta(z)$, but for a sufficiently flexible learned predictor they will be equal after LDM, and we don't make a difference here.

**Theorem 1 (restated).** *Assuming the following assumptions hold:*

*(i) (Data generation process) Data is generated by the predictive latent process Eq. (61) with invertible covariance $\Sigma$ and a differentiable and injective data generation function $g : \mathbb{R}^n \to \mathbb{R}^m$.*

*(ii) (Distribution matching) The model is defined through a sufficiently flexible encoder $f : \mathbb{R}^m \to \mathbb{R}^n$ to achieve perfect LDM (i.e., reach form 62), and $f$ is differentiable and invertible on the image of $g$, that is, $h = f \circ g$ is invertible.*

*(iii) (Predictor covers the latent space) The Jacobian $J_{p_R}(h(z_{:t}))$ has full row rank, equal to the latent dimension $n$ (and so does $J_p(z_{:t})$).*

*Then, distribution matching recovers the true latent variables up to affine transformations, that is, if the form of $R$ matches the form of $P$, the transforming function between latent variables is affine $h(z_t) = Az_t + c$.*

*Note* 1. We only require $h$ to be invertible, but not $f$ in general. This is important, since typically the data dimension is larger than the latent dimension $m > n$, making it impossible for $f$ to be invertible on the whole set $\mathbb{R}^m$.

*Note* 2. Assumption (iii) is required to prevent unconstrained dimensions in latent space in the last step of the proof. It is related, but not equivalent to the predictor being invertible—e.g., if the predictor depends only on the last step $p_R(z_{:t}) = p_R(z_{t-1})$, i.e., $J_{p_R}(h(z_{:t}))$ is square, then (3) implies $p_R$ is invertible, but for general predictors this assumption is a weaker constraint.

*Proof.* Since $R$ is completely defined by $h = f \circ g$ and $P$, and $h$ is invertible and differentiable (Assumptions i & ii), we can write $R$ based on the change of variables formula (note that we switch from $\tilde{z}$ to $z$ to simplify notation)

$$\log R(h(\boldsymbol{z})) + \log|J_h(\boldsymbol{z})| = \log P(\boldsymbol{z})$$
$$\log R(h(\boldsymbol{z})) + \sum_t \log|J_h(z_t)| = \log P(\boldsymbol{z})$$
$$\sum_t \log R(h(z_t)|h(z_{:t})) + \log|J_h(z_t)| = \sum_t \log P(z_t|z_{:t}) \, , \tag{63}$$

where the second line follows since $h(\cdot)$ is applied to the timesteps individually. We notice that the terms have to match per individual timestep. One way to see this is that, since the last representation $z_T$ only occurs once, the terms for $t = T$ have to match on both sides, and thus by induction (removing matching terms) all terms before. From this, and by using the form of the transformed latent distribution (62) we find

$$\log R(h(z_t)|h(z_{:t})) + \log|J_h(z_t)| = \log P(z_t|z_{:t})$$
$$-\log Z_R - \frac{1}{2}\|h(z_t) - p_R(h(z_{:t}))\|^2_{\Sigma_R^{-1}} + \log|J_h(z_t)| = -\log Z_P - \frac{1}{2}\|z_t - p(z_{:t})\|^2_{\Sigma^{-1}} \, , \tag{64}$$

where $Z$ are the normalization terms and we are using the weighted norm $\|x\|^2_A = x^T A x$. Finally, taking derivatives with respect to present and past latent variables, constant and single variable terms drop out and we find

$$\nabla_{z_t}\nabla_{z_{:t}}\left[\log R(h(z_t)|h(z_{:t})) + \log|J_h(z_t)|\right] = \nabla_{z_t}\nabla_{z_{:t}}\log P(z_t|z_{:t})$$
$$\nabla_{z_t}\nabla_{z_{:t}}\|h(z_t) - p_R(h(z_{:t}))\|^2_{\Sigma_R^{-1}} = \nabla_{z_t}\nabla_{z_{:t}}\|z_t - p(z_{:t})\|^2_{\Sigma^{-1}}$$
$$\nabla_{z_t}\nabla_{z_{:t}}\left[(h(z_t) - p_R(h(z_{:t})))^T\Sigma_R^{-1}(h(z_t) - p_R(h(z_{:t})))\right] = \nabla_{z_t}\nabla_{z_{:t}}\left[(z_t - p(z_{:t}))^T\Sigma^{-1}(z_t - p(z_{:t}))\right] \tag{65}$$
$$\nabla_{z_t}\left[-(J_{p_R}(h(z_{:t}))J_h(z_{:t}))^T\Sigma_R^{-1}(h(z_t) - p_R(h(z_{:t})))\right] = \nabla_{z_t}\left[-J_p(z_{:t})^T\Sigma^{-1}(z_t - p(z_{:t}))\right]$$
$$-(J_{p_R}(h(z_{:t}))J_h(z_{:t}))^T\Sigma_R^{-1}J_h(z_t) = -J_p(z_{:t})^T\Sigma^{-1} = \text{const. w.r.t. } z_t \, .$$

In the last line, since $J_{p_R}(h(z_{:t}))$ has full row rank (Assumption iii), and so have $J_h(z_{:t})$ ($h$ is invertible) and $\Sigma^{-1}$, it follows that $J_h(z_t)$ has to be constant w.r.t. $z_t$. A function with a constant Jacobian is necessarily affine, and we thus conclude that the transformation between true and recovered latent variables is an affine transformation $h(z_t) = Az_t + c$. □

*Note* 3. In equation 63 we assume that the in the first timestep $t = 0$ the uncoditional distributions match. Following the approach for conditional distributions, this could be ensured by assuming a Gaussian marginal distribution of $z_0$ and learning or enforcing the marginal. In practice it is sufficient to assume an empirical prior $P_\theta(z_0) = R(z_0)$ and hence ignore the first term, since recovery arises only from the conditional distributions.

The result is closely related to the Mazur-Ulam Theorem, which states that any surjective isometry (measure preserving map) between normed spaces is affine (Mazur & Ulam, 1932). Importantly, linearity results entirely from the noise structure of the Gaussian, which is equivalent to the norm of the space in Mazur-Ulam. This argument does not directly extend to variable covariance models—in the Kalman filter with input-dependent covariance, recovery likely stems from the constrained linear predictor.

The result generalizes the results of Zimmermann et al. (2021) and (most closely related) Laiz et al. (2025) under weaker assumptions. It does not only hold for a particular goal function, but for any algorithm that performs LDM while forcing the encoder to be injective. This could in principle also be performed via the reverse KL divergence, with the caveats mentioned in A.2.1, or other divergences. Further, the proof does not require any particular marginal distribution of the latents, such as uniform, Gaussian, or a learned correction.

Finally, the affinity of $h$ leads to a straightforward followup observation.

**Corollary 1.** *Under the assumptions of Theorem 1, and since the transformation function between latent variables is affine $h(z_t) = Az_t + c$, it follows that the learned predictor $p_R$ is an affine function of the true predictor $p$, i.e., $p_R(h(z_{:t})) = Ap(z_{:t}) + c = h(p(z_{:t}))$.*

*Proof.* The true conditional distribution is $z_t \sim \mathcal{N}(p(z_{:t}), \Sigma)$. Applying the affine transformation property of Gaussian random variables, the distribution of the transformed variable $h(z_t)$ is

$$h(z_t) \sim \mathcal{N}(Ap(z_{:t}) + c, A\Sigma A^T) . \tag{66}$$

By definition (62), the model parametrizes this distribution as $\mathcal{N}(p_R(h(z_{:t})), \Sigma_R)$. Matching the means of these two equivalent distributions yields

$$p_R(h(z_{:t})) = Ap(z_{:t}) + c = h(p(z_{:t})) . \tag{67}$$

$\square$

*Remark* 1. While these proofs concern the relation of latent variables $\tilde{z}_t = f(x_t)$ to true latent variables $z_t$, they do not tell us about the relation between the variables *within* the predictor. These are, for example, the 'hidden' latent variables $h_t$ of the Kalman filter (cf. Fig. 4), or the hidden variables in an RNN that are commonly used for downstream tasks (Oord et al., 2018). Here we remark that with a (partially) linear relation between 'hidden' and 'observed' latent variables ($h_t$ and $z_t$, respectively), and if the transformation function $h$ between true and recovered 'observed' latent variables is affine, then there exists a (partially) affine relation between true and recovered 'hidden' variables.

*Note* 4. Partially linear means that if we train the model based on a linear predictor $M$ from, e.g., $h_{t-1}$ to $z_t$, then only the row space of $M$ in $h_{t-1}$ is guaranteed to have a linear relation to $z_t$.

*Proof.* The composition of affine functions is affine. $\square$

Another straightforward insight is that theorem 1 can be extended to include models similar to CPC under the assumption that the MI term can be ignored (although even then CPC with the InfoNCE loss does not fully satisfy the assumptions, Aitchison & Ganev, 2024) with latent variables on the sphere.

**Theorem 2 (restated).** (Affine identifiability of vMF predictive model) *Under the assumptions (i)–(iii) of Theorem 1, and a von Mises-Fisher latent predictive distribution $P(z_t|z_{:t}) = \frac{1}{Z_P} \exp(\beta_P z_t^T p(z_{:t}))$ and a matching model, the transforming function $h$ between latent variables is affine.*

*Proof.* The proof remains the same until Equation 64, where we now find

$$\log R(h(z_t)|h(z_{:t})) + \log |J_h(z_t)| = \log P(z_t|z_{:t})$$
$$-\log Z_R + \beta_R h(z_t)^T p_R(h(z_{:t})) + \log |J_h(z_t)| = -\log Z_P - \beta_P z_t^T p(z_{:t}) . \tag{68}$$

After differentiation with respect to $z_t$ and $z_{:t}$ we are left with

$$\beta_R(J_{p_R}(h(z_{:t}))J_h(z_{:t}))^T J_h(z_t) = \beta_P J_p(z_{:t})^T = \text{const. w.r.t. } z_t \tag{69}$$

and the same conclusion follows. $\square$

### A.7.2. FOLDING ENTROPY

In this section we outline a formal proof of the folding entropy formula, in order to provide additional intuition. The formula holds under the following condition:

**Definition 3** (Piecewise invertible). An encoder is piecewise invertible on $\mathcal{M}$ if

$$f(x) = \begin{cases} f_1(x) & \text{if } x \in \mathcal{X}_1 \\ f_2(x) & \text{if } x \in \mathcal{X}_2 \\ \quad\vdots \end{cases} \tag{70}$$

and for all $\mathcal{X}_i \subset \mathcal{M}, x \in \mathcal{X}_i : f_i(x)$ invertible.

*Note* 5. For a differentiable (Lipschitz) encoder with nonzero singular values piecewise invertibility automatically holds. Since nonzero singular values are also encouraged by entropy maximization, and functions defined by neural networks are (mostly) differentiable, we can assume that during learning networks in LDM are piecewise invertible on the data manifold.

**Definition 4** (Folding entropy). For an encoder $f$ and distribution $P$ on $\mathcal{M}$, the *folding entropy* is

$$H_P[x|f(x)] := -\langle \log P(x|f(x))\rangle_{P(x)} , \tag{71}$$

where $P(x|f(x) = z)$ is the conditional distribution over preimages. The folding entropy satisfies $H_P[x|f(x)] \geq 0$, with equality if and only if $f$ is injective on $\text{supp}(P)$.

*Remark* 2. The folding entropy quantifies information lost due to non-injectivity: when $f$ maps distinct points to the same latent representation, uncertainty about the original point given its image increases. For injective $f$, the conditional satisfies $P(x|f(x)) = 1$ everywhere, so $H_P[x|f(x)] = 0$ and the formula reduces to the standard manifold change of variables. Inversely, if the folding entropy is nonzero $H_P[x|f(x)] > 0$ it captures information lost when $f$ collapses distinct points.

While the following formula has been proven previously by Geiger & Kubin (2012), we here give a different proof to provide intuition.

**Lemma 1** (Folding entropy decomposition). *Let $f : \mathcal{M} \to \mathbb{R}^n$ be a differentiable encoder with nonzero singular values, hence the Jacobian determinant of $f$ on $\mathcal{M}$ is non-zero $|J_f(x)J_f(x)^T|^{1/2} > 0$ (required to avoid infinities). Let $R(z) = \langle \delta(z - f(x))\rangle_{P(x)}$ be the pushforward of $P$ under $f$. Then*

$$H_R[z] = H_P[x] + \left\langle \frac{1}{2}\log|J_f(x)J_f(x)^T| \right\rangle_{P(x)} - H_P[x|f(x)] . \tag{72}$$

*Proof.* We first find an expression of the latent distribution $R(f(x))$. Define the local volume-adjusted density $r(x) := P(x)/|J_f(x)J_f(x)^T|^{1/2}$, which accounts for the change of volume element under $f$ (Krantz & Parks, 2008, p. 125). Given the assumption of piecewise invertible $f$, the pushforward density at $z \in f(\mathcal{M})$ is

$$R(z) = \sum_{x \in f^{-1}(z)} r(x) , \tag{73}$$

where the sum runs over all preimages of $z$. The conditional distribution over preimages is then the local density normalized by the contributions of all preimages

$$P(x|f(x)) = \frac{r(x)}{Z} = \frac{r(x)}{\sum_{x' \in f^{-1}(f(x))} r(x')} = \frac{r(x)}{R(f(x))} . \tag{74}$$

Rearranging gives $R(f(x)) = r(x)/P(x|f(x))$, and substituting the definition of $r(x)$ we get the desired expression

$$R(f(x)) = \frac{P(x)}{|J_f(x)J_f(x)^T|^{1/2}P(x|f(x))} . \tag{75}$$

We can compute the latent entropy by first taking logarithms and then taking the average with respect to $x$

$$\log R(f(x)) = \log P(x) - \frac{1}{2}\log|J_f(x)J_f(x)^T| - \log P(x|f(x)) , \tag{76}$$

leading to

$$H_R[z] = -\langle \log R(f(x)) \rangle_{P(x)} \tag{77}$$

$$= -\langle \log P(x) \rangle_{P(x)} + \left\langle \frac{1}{2} \log |J_f(x) J_f(x)^T| \right\rangle_{P(x)} + \langle \log P(x|f(x)) \rangle_{P(x)}$$

$$= H_P[x] + \left\langle \frac{1}{2} \log |J_f(x) J_f(x)^T| \right\rangle_{P(x)} - H_P[x|f(x)] , \tag{78}$$

where we identified $-\langle \log P(x) \rangle_{P(x)} = H_P[x]$ and $-\langle \log P(x|f(x)) \rangle_{P(x)} = H_P[x|f(x)]$. $\qquad\square$

### A.8. Investigation of categorical model with probabilistic encoder

We also wanted to understand if LDM in latent space allows for uncertainty in representations of single observations. To this end we used the same non-temporal goal functions as before, but defined a recognition distribution $R(z|x)$ that is not a simple delta peak.

**Categorical model**  To enable analytical tractability, we defined a simple categorical model that assigns individual probabilities to $n$ categories via a softmax encoder We make the choices (with $n$ categories $k$)

$$R(z = k|x) = p_k(x) = \frac{1}{Z(x)} \exp(f_k(x))$$

$$P_\theta(z = k) = \frac{1}{N} \tag{79}$$

$$P_\theta(z' = k'|z = k) = \frac{1}{Z} \exp(\beta \mathbb{1}[k' = k]) .$$

This model assumes that if $z$ and $z'$ are encodings of related samples, they are assigned to the same category with probability $p_\theta = \exp(\beta)/(\exp(\beta) + n - 1)$. The likelihood term becomes

$$\left\langle \sum_{k=1}^{n} \sum_{k'=1}^{n} \log P_\theta(z = k|z' = k') R(z = k|x) R(z' = k'|x') \right\rangle_{P_{\text{data}}(x,x')}$$

$$\propto \left\langle \sum_{k=1}^{n} \sum_{k'=1}^{n} \beta \mathbb{1}[k' = k] \frac{1}{Z(x)} \exp(f_k(x)) \frac{1}{Z(x')} \exp(f_{k'}(x')) \right\rangle_{P_{\text{data}}(x,x')} \tag{80}$$

$$= \beta \left\langle \sum_{k=1}^{n} p_k(x) p_k(x') \right\rangle_{P_{\text{data}}(x,x')} .$$

The entropy term can be computed analytically

$$H_R[z] = -\sum_{k=1}^{n} R(z = k) \log R(z = k)$$

$$\approx -\sum_{i} \sum_{k=1}^{n} \frac{1}{Z(x_i)} \exp(f_k(x_i)) \log \left[ \sum_{j} \frac{1}{Z(x_j)} \exp(f_k(x_j)) \right] . \tag{81}$$

This results in the goal function

$$\mathcal{F} = \beta \sum_{k=1}^{n} \langle R(z = k|x) R(z' = k|x') \rangle_{P_{\text{data}}(x,x')} + H_R[z, z'] . \tag{82}$$

**Simulations**    We tested the ability of the model to learn probabilistic representations on MNIST, by providing two different images of the same digit as input to the SSL algorithm. We assumed $n = 10$ categories, and that two presented digits can be detected to be in the same category with probability $p_\theta = 0.99$ by choosing $\beta$ in the model accordingly (Fig. A8A). To prevent model collapse in the beginning of learning, we annealed $p_\theta$ over epochs to the final value, starting from $p_\theta = 0.8$. After learning and sorting learned categories, the algorithm recovered the digit identity near perfectly with an accuracy of $0.99$ (Fig. A8B). In contrast, with additional MI maximization the model would collapse categories under the correct latent model. Instead, we had to assume a misspecified latent model with $p_\theta = 0.8$ that effectively counteracts MI maximization to achieve reliable representation learning of digit identities.

We also found that representations with $\mathcal{F}_{\text{LDM}}$ had meaningful quantification of uncertainty: Digits that were assigned to a single category (low entropy encodings) were consistently highly stereotypical, while digits with uncertain encoding (high entropy) were outliers without clear identity (Fig. A8C,D). In comparison, with additional MI maximization, the algorithm predominantly forms extremely low-entropy representations (Fig. A9). Evidently, here MI is not maximized through the presence of an invertible encoder, and thus plays a non-neglegible role in shaping representations.

LDM is able to produce probabilistic encodings that respect epistemic uncertainty, which has previously been demonstrated by Kirchhof et al. (2023). Notably, while Kirchhof et al. (2023) employed the reverse KL divergence, which requires reparametrization sampling to evaluate the goal function, the approach here is simpler and does not require sampling. Crucially, however, the motivation through the data likelihood is based on the assumption of invertible encoders, which clearly conflicts with probabilistic encoders. Indeed, in preliminary experiments we found that it is not easily possible to optimize model parameters $\theta$ in the same way as it is for deterministic encoders, which likely is caused by this mismatch. A theoretical framework for how flexible probabilistic models based on latent space prediction are related to LDM might be provided via more complex theoretical approaches in the future, e.g., recognition parametrized models (Walker et al., 2023; Hromadka et al., 2025).

# B. Simulation details

Details on parameters and precise implementation can be found in the simulation code at https://github.com/fmi-basel/latent_distribution_matching.

**Learning of image representations.** For experiments we extended the `solo-learn` library provided by da Costa et al. (2022), which uses ResNet-18 as standard encoder and 1000/400 epochs of training for CIFAR/Imagenet-100. Note that for Imagenet-100 we used the classes as used by Tian et al. (2020), which were different than those used by da Costa et al. (2022). For latents on the plane we used a Gaussian conditional model as described in A.6.1; for latents on the sphere we used a vMF conditional model as described in A.6.1. We implemented the entropy estimators as described in A.5, with the following details. For contrastive entropy estimation via KDE (A.5.1) we used vMF kernels on the sphere, leading to the same loss as in Chen et al. (2020), and Gaussian kernels on the plane. For non-contrastive entropy estimation (A.5.3) on the plane, for comparability, we used the variance-covariance regularization as originally proposed in Bardes et al. (2021). On the sphere we used the log-det expansion from A.5.3. For kNN entropy estimation (A.5.2) we used the euclidean metric ($p = 2$), chose $k = 3$ and discarded the upper 10% of kNN, which we considered outliers and already well separated.

**Learning of a simple dynamical system with Kalman-based SSL.** We randomly sampled 800 training sequences consisting of trajectories and noise and trained for 20 epochs—note that only the simulations in Fig. A4 used noise in the trajectories. Synthetic videos of 10 by 10 pixel resolution were then created according to the process outlined in the main text. We specified the encoder through a simple MLP with ReLU activation and one hidden layer with 100 units. We specified the Kalman filter with diagonal covariances $\Sigma_A$ and $\Sigma_D$. $D$ was specified, without loss of generality, as a fixed diagonal matrix, with or without zeros on the diagonal, depending on whether 'hidden' latent states were desired or not. For both models using stopgrad (A.6.2) and kNN (A.5.2) entropy estimation, we learned the Kalman filter on faster timescales than the encoders. For kNN entropy estimation, again, we used the euclidean metric ($p = 2$), chose $k = 3$ and discarded the upper 10% of kNN, which we considered outliers and already well separated.

**Modeling hippocampal activity dynamics with Kalman-based SSL.** We modeled the data of the `hc-11` dataset (Grosmark & Buzsáki, 2016) based on the pre-processing as described in Schneider et al. (2023), i.e., model inputs were spike-counts of 120 neurons in 25 ms bins. For one epoch we sampled 100000 windows of length $\sim$10 s from the full spike-train (about 45 min), training for 16 epochs. For the Kalman filter we chose an 8D observation and 16D latent space. For the encoders we used the same setup as before. We used kNN entropy estimation, with the same parameters as before. For a linear position predictor the prediction distribution can be computed analytically. To find 95% confidence intervals for MLP position prediction, for each timestep we produced 1000 samples of the latent state distribution $\mathcal{N}_{h_t}(\mu_{h_t}, \Sigma_{h_t})$ as given by the Kalman filter backbone. We then computed the predicted positions for these samples through the MLP (trained by predicting position from latent mean $\mu_{h_t}$), and found the 95% sampling based confidence intervals.

**Nonlinear systems identification with predictive distribution matching.** We randomly sampled 10000 sequences and trained for 300 epochs (although convergence was observed much earlier). The encoder was specified as an MLP with ReLU activation, 10 hidden layers, and 200 hidden units per layer. The predictor, mapping from all previous 2 dimensional encodings to the next encoding, was an LSTM RNN with an MLP predictor head. The LSTM had 1 layer with 10 hidden units. The predictor head was an MLP with ReLU activation, 2 hidden layers, and 20 hidden units. In the main text we used kNN entropy estimation, with the same parameters as before. In the Appendix Fig. A6 we used the stopgrad entropy estimator approximation as defined before, the LogDet entropy estimator with the `slogdet` function in pytorch, and the KDE estimator with bandwidth 1. For the second experiment (Fig. A7) we used the same setup, training on 100000 samples for 20 epochs.

**Probabilistic representations in categorical SSL.** We used ResNet-18 as encoder, with an additional soft-max output layer, training for 30 epochs.

## C. Supplementary figures and tables

*Table A1.* Validation accuracy for linear probing on CIFAR-10, CIFAR-100, SVHN, and Imagenet-100. We used a standard SSL image pretraining setup as described by da Costa et al. (2022). *MI max.* denotes whether $\mathcal{F}_{\mathrm{LDM}}$ (○) or $\mathcal{F}_{\mathrm{MI}}$ (●) was used. Standard deviation was calculated based on three runs. Note that Imagenet-100 classes were selected according to Tian et al. (2020), which were different than those used by da Costa et al. (2022), accounting for the difference in performance. The implementation of VICReg (Plane-LogDet-●) is the same as in da Costa et al. (2022) and serves as a reference.

| | Model | | CIFAR-10 Acc. | | CIFAR-100 Acc. | | SVHN Acc. | | Imagenet-100 Acc. | |
| Space | Entropy est. | MI max. | Top-1 | Top-5 | Top-1 | Top-5 | Top-1 | Top-5 | Top-1 | Top-5 |
|---|---|---|---|---|---|---|---|---|---|---|
| Plane | Contr. | ○ | 91.87±0.06 | 99.73±0.00 | 65.28±0.28 | 89.06±0.24 | 92.15±0.17 | 99.07±0.13 | 72.60 | 92.36 |
| Plane | Contr. | ● | 92.09±0.08 | 99.76±0.04 | 65.29±0.15 | 88.96±0.25 | 92.20±0.11 | 99.11±0.03 | 72.68 | 92.04 |
| Plane | kNN | ○ | 92.07±0.17 | 99.68±0.05 | 65.63±0.36 | 88.93±0.06 | 92.75±0.13 | 99.20±0.04 | 74.32 | 93.36 |
| Plane | kNN | ● | 91.89±0.05 | 99.69±0.03 | 65.85±0.38 | 88.94±0.17 | 92.85±0.11 | 99.19±0.04 | 73.68 | 92.84 |
| Plane | LogDet | ○ | 92.06±0.05 | 99.74±0.02 | 69.47±0.21 | 91.27±0.15 | 92.79±0.13 | 99.20±0.04 | 75.92 | 93.32 |
| Plane | LogDet | ● | 91.94±0.05 | 99.74±0.02 | 68.62±0.06 | 90.77±0.19 | 92.72±0.09 | 99.13±0.04 | 74.72 | 93.40 |
| Sphere | Contr. | ○ | 90.93±0.13 | 99.74±0.03 | 64.83±0.19 | 88.63±0.21 | 91.59±0.05 | 98.92±0.05 | 72.12 | 92.12 |
| Sphere | Contr. | ● | 91.38±0.22 | 99.73±0.03 | 66.01±0.24 | 89.47±0.14 | 92.33±0.12 | 99.05±0.04 | 73.08 | 92.72 |
| Sphere | kNN | ○ | 90.22±0.04 | 99.68±0.02 | 64.31±0.07 | 88.00±0.09 | 92.27±0.12 | 98.99±0.04 | 72.64 | 92.16 |
| Sphere | kNN | ● | 89.96±0.16 | 99.63±0.04 | 64.50±0.18 | 87.82±0.12 | 92.16±0.03 | 99.05±0.02 | 73.28 | 92.28 |
| Sphere | LogDet | ○ | 91.41±0.21 | 99.77±0.02 | 65.40±0.18 | 89.31±0.11 | 92.57±0.02 | 99.09±0.03 | 73.00 | 93.00 |
| Sphere | LogDet | ● | 91.20±0.11 | 99.73±0.01 | 65.37±0.16 | 89.34±0.22 | 92.54±0.04 | 99.09±0.01 | 72.28 | 92.40 |
| Simplex | Contr. | ● | 90.23±0.13 | 99.69±0.03 | 62.65±0.22 | 86.98±0.12 | 92.21±0.07 | 99.03±0.05 | 70.50 | 91.97 |

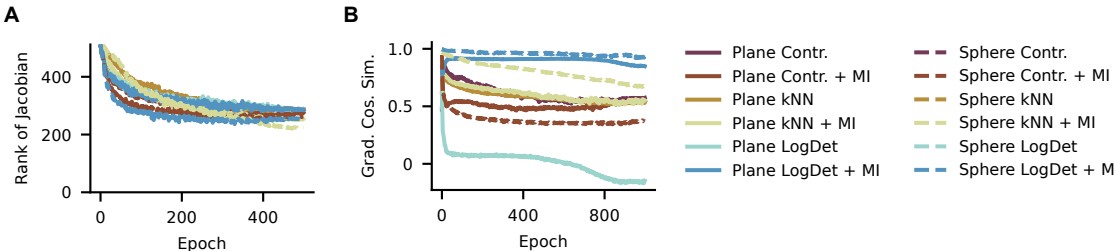

*Figure A2.* **A** The rank of encoder Jacobian reaches similar levels for all approaches (here, averaged over data-points on CIFAR-10). The rank is regularized by the entropy term, since without the entropy it collapses to zero. Note that even if the final rank exceeds the dimensionality of the data manifold (in this case estimated to be less than 100 dimensions) this does not imply local invertibility. However, we tested for local invertibility on a toy dataset with known data generation process (detailed in Fig. A7), by computing the rank of the combined generator-encoder function. This function was consistently full rank, implying local invertibility. **B** Similarity of single and joint entropy gradients of $\mathcal{F}_{\mathrm{MI}}$ and $\mathcal{F}_{\mathrm{LDM}}$, respectively, for CIFAR-100. Single and joint entropy gradients point into consistent directions throughout learning, with the exception of the LogDet estimator on the plane for $\mathcal{F}_{\mathrm{LDM}}$. That is, when training with $\hat{H}[z, z']$ in $\mathcal{F}_{\mathrm{LDM}}$, the estimated entropy $\hat{H}[z, z']$ has slightly opposing gradient to $\hat{H}[z]$ towards the end of the training (but not the other way around when training with $\mathcal{F}_{\mathrm{MI}}$). The likely reason are the strong assumptions about the shape of the latent distribution by the LogDet estimator.

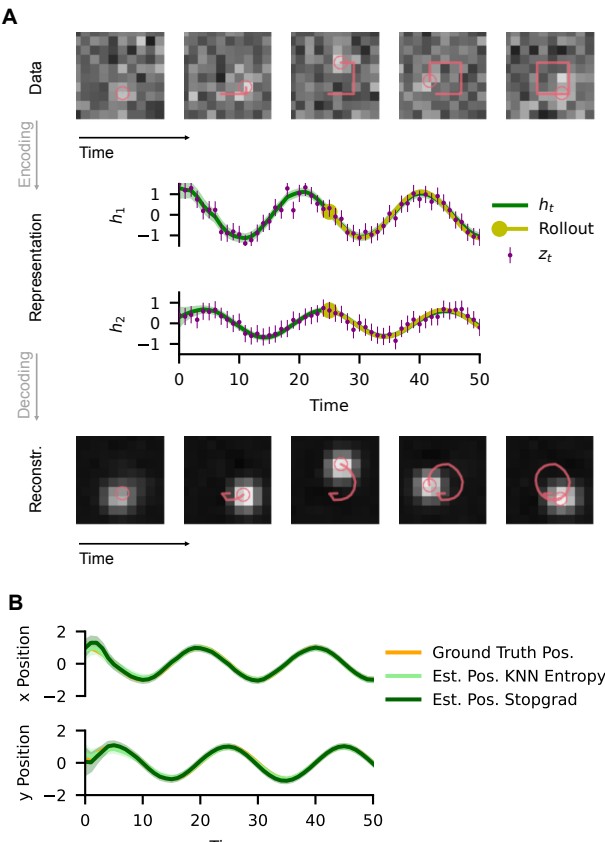

*Figure A3.* Square movement task. **A** A dot moving on a square trajectory is learned to be mapped to a linear dynamical system. The learned system captures the inferred dynamics, demonstrated by simulating a latent trajectory based on the latent state at $t = 25$. We also show the reconstructed input (MLP decoder) and trajectory (linear decoder). Linear decoding does not enable decoding the correct trajectory. **B** To estimate decoding performance and uncertainty based on a linear decoder, we provide a surrogate true position, which is a sinusoidal with the frequency of the moving dot. Both contrastive and stopgrad predictive SSL find good encodings $R^2 = 0.97$.

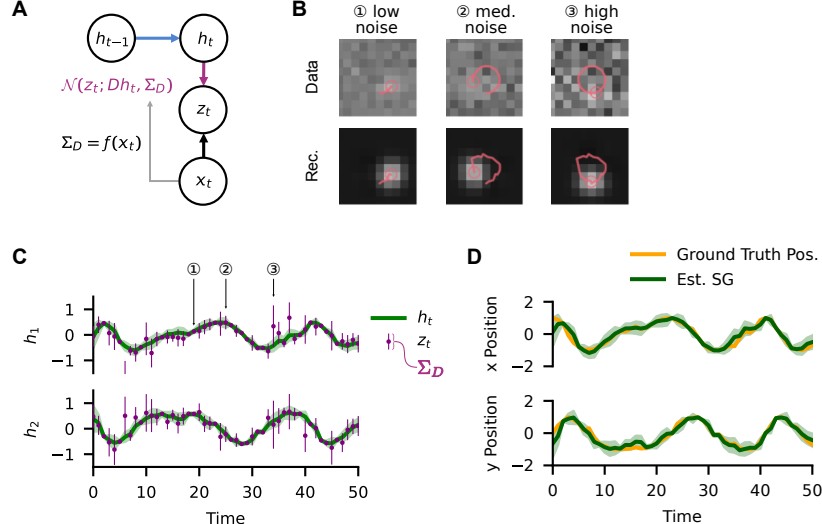

*Figure A4.* Kalman SSL with input dependent observation noise. **A** Observation noise (diagonal $\Sigma_D$) is estimated with an MLP with softplus head. **B** Data has randomly chosen noise level every timestep. **C** The model correctly infers the changing noise level and **D** finds a good estimate of the latent trajectory in the hidden variables $h_t$ ($R^2 = 0.96$).

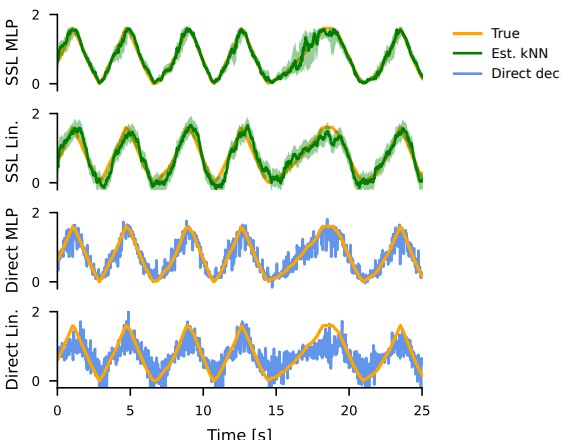

*Figure A5.* Comparison of position prediction through Kalman-based SSL or directly from the data. Kalman-based SSL makes highly accurate predictions with MLP decoder ($R^2 = 0.98$). With a linear decoder prediction becomes worse since at the turning points real position dynamics are not well approximated by a linear model ($R^2 = 0.88$, similar problem as in Fig. A3). For direct prediction we use the spikes binned in 25ms time windows, which leads to inferior performance of MLP ($R^2 = 0.88$) and Linear ($R^2 = 0.57$) predictors.

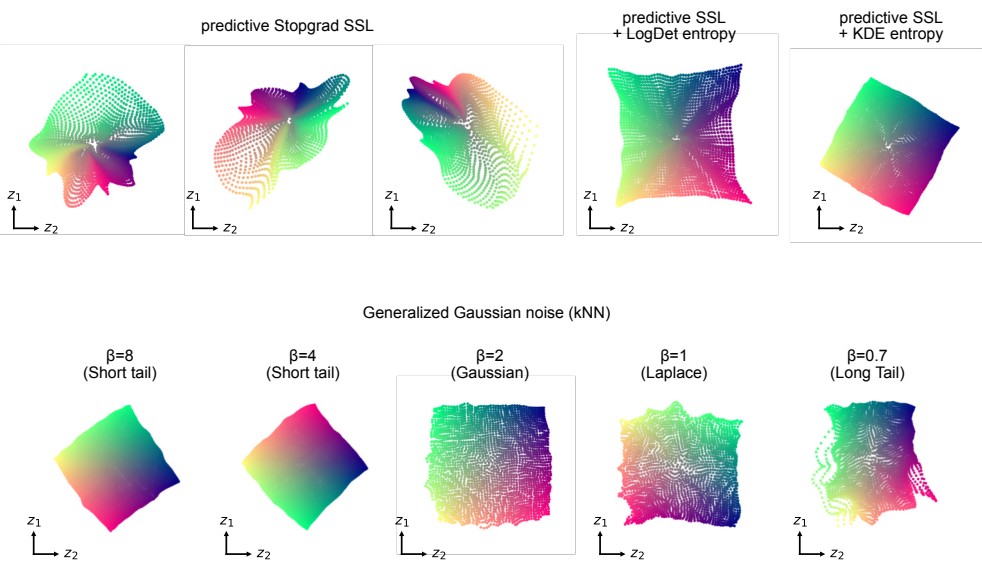

*Figure A6.* **Top** stopgrad based predictive SSL leads to nonlinear identification of the underlying dynamics, which we exemplify with three learning outcomes (cf. Fig. 5C). The resulting data representations are approximately locally linearlized on the latent plane, and learned dynamics are predictive of future representations (not shown), but, since stopgrad based predictive SSL employs a heuristic entropy estimator (which already assumes the conditional distribution to be Gaussian), they retain nonlinear distortions compared to the true variables. Also, when using the LogDet entropy estimator (instead of kNN in the main paper) affine identification is only approximate. The reason is that for the underlying variables of this particular data the Gaussian assumption of the LogDet estimator does not hold, which thus leads to a biased entropy estimate. Such biased entropy estimates, such as in VICReg (Bardes et al., 2021) or SICReg (Balestriero & LeCun, 2025), could also be a problem in practice, even when exact identification is not required, since it can lead to a mismatch between empirical and model conditional distributions (e.g., the model makes Gaussian-distributed predictions, while the empirical conditional latent distribution has a different shape). For KDE entropy estimation we find similarly good results as for the kNN estimator. **Bottom** Identifiability is robust to limited deviations from Gaussian noise in the conditional distribution of the data generation process. Here we replaced Gaussian noise with Generalized Gaussian noise with long or short tails. Approximate identification persisted for non-isotropic noise as displayed here (notice also the alignment of the recovered manifold with the axes). For isotropic noise, or for multi-modal conditional distributions, we did not observe the same robustness in general (not shown), suggesting that more complex latent models are needed in such cases.

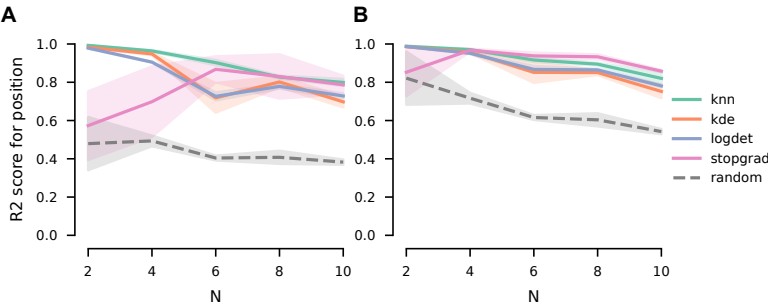

*Figure A7.* Higher-dimensional nonlinear source recovery task. We follow standard practice in identifiability literature (e.g. Khemakhem et al., 2020b), with a nonlinear latent relation and an invertible DNN based generation function. We sample two latent variables $z$, $z'$, where $z \sim \mathcal{N}(0, \mathbb{1})$ and $z' \sim \mathcal{N}(\mu(z), \alpha\mathbb{1})$, $\mu(z) = Rz + 0.1 \tanh(Rz)$, $R$ is a random rotation matrix. 100-dimensional observations are generated from $z$ and $z'$ via an invertible (LeakyReLU) random 3-layer MLP. **A** Recovery generally works well in low dimensions $N$ (except for stopgrad models) and deteriorates for $N > 4$. We suspect two reasons: 1) Higher $N$ require exponentially more samples to cover the latent space (for intuition, the $10^5$ samples in $N = 10$ dimensions correspond to roughly 3 points per axis in a grid); 2) entropy maximization becomes more challenging, since 'unfolding' the latent distribution requires pushing the 'creases' through the distribution itself. This process slows down when there are more creases and dimensions. Still, recovery consistently outperforms randomly initialized models. **B** To test the second hypothesis, we trained overcomplete models with latent dimensionality $M = 2 \cdot N$, which allows probability mass to 'slip past' itself more easily. We find improved recovery and more reliable training for all models, even though now representations could in principle be nonlinearly related to true latents due to the mismatch in dimensionality. Shaded regions denote 95% confidence intervals based on standard error for 5 runs.

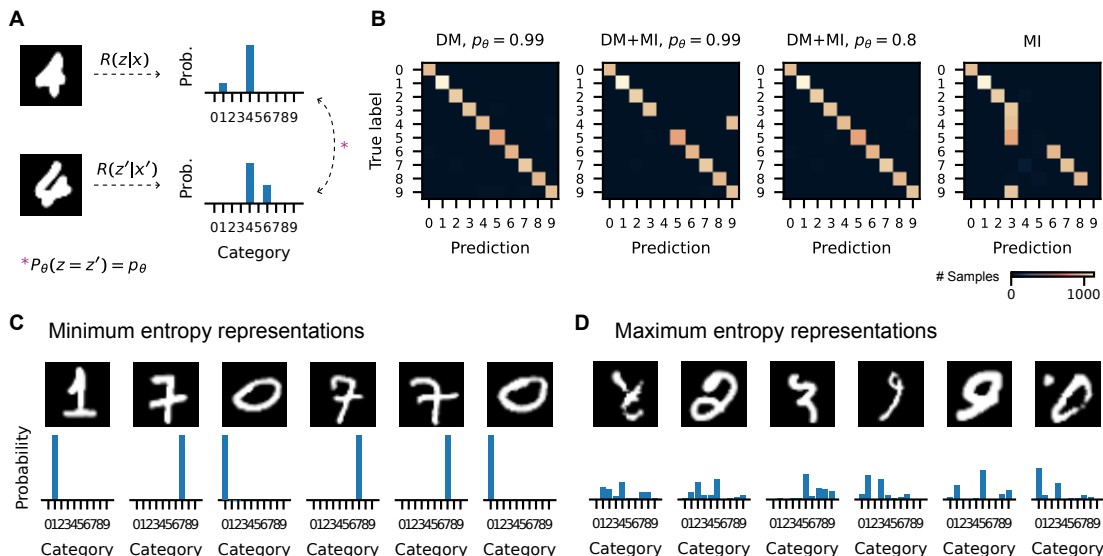

*Figure A8.* SSL based probabilistic representations of MNIST digits. **A** After digits are encoded with softmax encoders $R(z|x)$, the algorithm encourages them to be in the same category with probability $p_\theta$. **B** LDM recovers digit identities after sorting of categories with an accuracy of 0.99. With MI maximization the model has to be mis-specified (low matching probability of $p_\theta = 0.8$) to prevent collapse of digit identities. MI as an optimization goal alone shows poor results. **C** Most certain digit encodings. **D** Most uncertain digit encodings.

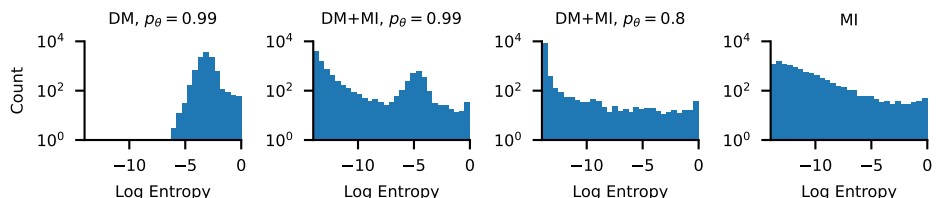

*Figure A9.* Histogram of entropies in digit encodings. LDM alone leads to moderate to high uncertainty in digit encodings, while (additional) MI maximization results in extremely low estimated uncertainties.

