# OpenReview forum: "Understanding Self-Supervised Learning via Latent Distribution Matching"
_ICML.cc/2026/Conference — ICML 2026 spotlight_

### Official Review · Reviewer_WDsE · 2026-03-11

**Soundness:** 3
**Presentation:** 3
**Significance:** 3
**Originality:** 3
**Overall Recommendation:** 5
**Confidence:** 3

**Summary:**

The paper unifies Distribution matching with contrastive, noncontrastive, and predictive SSL methods, including stop-gradient approaches. Leveraging the view from DM, the paper derive a nonlinear, sampling-free Bayesian filtering model with a Kalman-based predictor for high-dimensional timeseries. Empirical evidence shows that many SSL methods indeed show no big difference between the unifying framework and its original form. Theoretical hypothesis are also proposed to support the claims.

**Compliance With Llm Reviewing Policy:**

Affirmed.

**Final Justification:**

The authors have clarified all of my questions. I therefore raised my score.

**Key Questions For Authors:**

Please see weakness above for the questions to be addressed during rebuttal. Please also correct me if I misunderstood anything.

**Limitations:**

Yes, they authors did.

**Strengths And Weaknesses:**

Strength:
1.	The paper casts different SSL methods as latent distribution matching (DM): learning representations that maximize log-probability while maximizing latent entropy to prevent collapse. Leveraging DM, the authors derive a nonlinear, sampling-free Bayesian filtering model with a Kalman-based predictor for high-dimensional timeseries.

2.	To demonstrate how DM enables deriving new SSL algorithms, the paper illustrates how to consider a model in which the predictor given by a Kalman filter as an example, and shows that after learningof the filter, both CPC (with MI maximization) and predictive stopgrad SSL (without MI maximization) recovered the underlying latent variables equally well, verifying the assumption that SSL and be unified throught the lens of DM.

3.	Theoretical guarantees are provided to support the claims above with mild assumptions.


Weakness:
1.	The empirical evidences are only implemented on small datasets of Cifar and Imagenet100, which are small. While self-supervised learning is known to only work for large pre-training dataset, the current setup may have weakened the evidence supporting the unifying claims in the paper.  Can authors consider imagenet1K?
2.	Could the authors please further clarify the goal of the Kalman filter section. I understand that this is one example that illustrates how the proposed unification helps temporal model, but after showing the link, there is no empirical evidence further supporting the usefulness of the unification assumption in the Kalman filter scenario, leading to this paragraph standalone.
3.	Section Empirical validation (7.1) only considers 2 dimensional case. This is very small , can the authors consider larger latent dimension for synthetic data and compare the latent prediction accuracy ?

---

> ### Author Rebuttal · Authors · 2026-03-31
>
> 1. (Limited evaluation) We agree that empirical validation of the claims is important. We will therefore include additional empirical tests on larger and more diverse datasets. We have already run additional experiments on SVHN and the results are consistent with what we have presented. We aim to include larger datasets into the final revision of the paper, but we're not sure imagenet1k is feasible with our current resources.
>
> 2. (Goal of Kalman filter) The goal of the Kalman filter section is twofold. On the one hand, it provides a concrete example of a novel SSL algorithm derived from our DM framework. On the other hand, this novel SSL approach has practical advantages over previous SSL approaches because it gives uncertainty estimates over the inferred latents for free.  To illustrate this in a more realistic scenario, we turned to Hippocampal activity recordings of rats running on a linear track (hc11 on CRCNS.org, see also Reviewer 1 question A for more details). The key point is that the proposed approach learns latent representations together with analytical confidence estimates on the latent states. This allows us to find precise decodings of the rat position from latent representations of hippocampal activity (similar to previous SSL approaches), and supplement these estimates with confidence intervals computed from the Kalman filter (novel to our approach). We think this is a neat application of our DM framework and we will make sure to better explain these key contributions in the revised paper.
>
> 3. (Empirical validation of identifiability in high dimensions) Yes, we have now investigated this, thanks for asking. To look at higher dimensional latent spaces, we designed a new synthetic dataset that allows us to control the actual latent and observation dimensions, following the standard practice in the identifiability literature (e.g., Khemakhem et al., 2020). Concretely, we consider two related "true" D-dimensional samples z1 and z2. z1 is a zero-mean Gaussian variable, and z2 is obtained by applying a nonlinear invertible transformation to z1 and adding Gaussian noise. Both samples are then mapped to high dim observation space through random, invertible MLPs. Initial experiments (with the same network setup as before in Figure 5) show promising results, with high R2 for low dimensional latent spaces up to D=4. With higher dimensional latent spaces we see a gradual drop-off in identifiability for the short training runs we conducted. We speculate this effect is related to optimization difficulties, due to two reasons: i) Higher dimensions require exponentially many samples to achieve similar coverage, slowing down convergence significantly; ii) Identification requires to "unfold" the true latent space in the recovered latent space, which means moving the "creases" through the probability mass itself, which we suspect to become much slower with high D. To further investigate the latter idea (ii) we relaxed the requirement of having equal dimensionality of true and recovered latent space and enlarged the dimensionality of the recovered space. This allows the probability mass to effectively "slip past" itself, and consistent with our hypothesis, we observed a significant increase in the linear predictability of true latent variables in higher D. We also checked whether the increased performance is simply due to the increased latent dimensionality, which would improve linear readout already for random projections, but this is not the case. We will continue investigating this interesting setting and will include these findings to the final manuscript.

---

> > ### Author Rebuttal · Reviewer_WDsE · 2026-04-01
> >
> > Thanks for the clarification to my questions. I therefore raise my score from weak accept to accept.

---

### Official Review · Reviewer_B8z9 · 2026-03-12

**Soundness:** 3
**Presentation:** 3
**Significance:** 3
**Originality:** 2
**Overall Recommendation:** 5
**Confidence:** 3

**Summary:**

This paper presents a theoretical framework for self-supervised learning (SSL) based on latent variable distribution matching. It unifies a variety of SSL methods under the lens of minimizing the KL divergence between the distribution of augmented data and a hypothesized latent variable model. The framework decomposes latent variable distribution matching into an alignment term and a uniformity term, yielding an intuitive formulation. Under certain assumptions, the authors also prove that the learned latent space can recover the true latent variables up to an affine transformation.

**Compliance With Llm Reviewing Policy:**

Affirmed.

**Final Justification:**

The rebuttal addressed my main concerns about originality. I have decided to raise the score from weak accept to accept.

**Key Questions For Authors:**

The originality is questionable -- what is really new compared with, say I-con? Anything tangible to claim and explain here?

**Limitations:**

The paper indeed discusses its limitations.

**Strengths And Weaknesses:**

Strengths
Soundness: The derivation of the unified framework is clear. The experimental design, along with the newly proposed Kalman-based predictor for time series, provides reasonable support for the paper's central claim that entropy estimation is more critical than mutual information.

Significance: The topic of understanding SSL is both interesting and important, and remains an area that warrants further development. The distribution matching framework proposed here makes a solid contribution toward that goal.

Weaknesses
Originality: I am not yet clear on what advantages this work has over I-con. I would appreciate it if the authors could provide a more concrete and formal comparison, particularly in terms of mathematical framework designs and general applicability.

---

> ### Author Rebuttal · Authors · 2026-03-31
>
> Thanks for this question. We realize that we did not sufficiently delineate how our work differs from the I-con paper. There are several important differences. While I-con also minimizes a KL divergence, it operates on conditional neighborhood distributions over data points - aligning a supervisory distribution $p(j|i)$, i.e., the assumed relationships between samples, with a learned distribution $q(j|i)$ derived from the representations. In contrast, our DM framework directly matches distributions in latent space rather than over data points - aligning the 'empirical' latent distribution to a model distribution. This seemingly subtle difference in which distributions are being matched leads to profound differences in the derivations of the SSL loss functions with several consequences. On the one hand, it is not clear how identifiability guarantees could be derived within the I-con framework, whereas this is possible within our DM framework, as we show. On the other hand, I-con does not yield explanations why the plethora of previously proposed MI-based approaches have been successful for representation learning---which our DM framework does provide. Finally, it is not clear how one would derive temporal models in I-con, such as the Kalman-filter based approach in our paper. We will clarify the difference between the frameworks in the manuscript.

---

> > ### Author Rebuttal · Reviewer_B8z9 · 2026-04-04
> >
> > The distinction from I-con was clearly addressed; the theoretical framework does differ substantially. I have decided to raise the score from weak accept to accept.

---

### Official Review · Reviewer_3gzH · 2026-03-12

**Soundness:** 3
**Presentation:** 3
**Significance:** 3
**Originality:** 3
**Overall Recommendation:** 4
**Confidence:** 3

**Summary:**

This paper proposes a unifying theory of self-supervised learning by casting SSL as latent distribution matching (DM). Under this view, SSL learns representations that maximize their log-probability under an assumed latent model while maximizing latent entropy to avoid collapse, thereby connecting alignment and uniformity to a more formal statistical framework. The paper claims that this perspective unifies contrastive, non-contrastive, predictive, and stop-gradient approaches, derives a predictive SSL model with a Kalman-based latent dynamics component, and proves identifiability of predictive DM up to affine transformations under mild assumptions.

**Compliance With Llm Reviewing Policy:**

Affirmed.

**Key Questions For Authors:**

Can the authors delineate more precisely which SSL families are exactly derivable from latent DM and which only admit an approximate interpretation?

What are the strongest assumptions required for identifiability in the predictive setting?

Can the authors add broader empirical comparisons showing the explanatory power of DM across representative contrastive, non-contrastive, and stop-gradient methods in a common evaluation protocol?

**Strengths And Weaknesses:**

Strengths.

Rather than proposing one more SSL objective, it tries to organize a large and somewhat fragmented literature under a common latent distribution matching principle. That kind of unification can be very valuable if done carefully.

The paper argues that mutual-information maximization alone is insufficient as an explanatory principle because MI is invariant under invertible transformations, whereas distribution matching can support identifiability under appropriate assumptions. It then uses this perspective to reinterpret existing SSL methods and to motivate new predictive objectives. This is a substantial theoretical narrative rather than a superficial relabeling.

The theory leads to a constructive model. The paper does not stop at reinterpretation: it derives a predictive SSL model with Kalman-based latent dynamics and emphasizes principled uncertainty quantification. This makes the contribution more than a survey-like unification.

The identifiability discussion is particularly promising. The paper explicitly claims that predictive DM can recover latent variables up to affine transformations, and the nonlinear-system-identification experiment shown later is consistent with that claim, including the observation that the recovered space becomes injective after entropy-driven learning.

Weaknesses:

The paper claims a fairly broad unification of SSL, including stop-gradient methods, but broad unification claims are easy to overstate. I would like a crisper account of which families are fully captured by the DM framework and which are only approximately interpreted through it.

The empirical section, based on the material available here, seems lighter than the theoretical ambition. For a paper making strong claims about the principles underlying SSL, I would ideally want a larger and more systematic empirical demonstration across major SSL families, rather than mainly representative examples and a nonlinear system-identification study.

Some claims also need sharper operationalization. For example, the statement that approximate MI maximization implicitly performs DM is intriguing, but it would be helpful to see more direct empirical evidence that tracks this correspondence across objectives rather than relying mainly on conceptual argument.

---

> ### Author Rebuttal · Authors · 2026-03-31
>
> Q1: Which SSL families are exactly derivable from latent DM?
>
> Thanks for this question, we recognize that we should have been more explicit about this point.
> We showed that we can directly derive classic contrastive (SimCLR), regularization-based (VICReg), and stopgrad SSL approaches (SimSiam) within our DM framework. However, our framework is more general than that. The most general characterization: Any latent graphical model of the form $\prod_i P_\theta(z_i|PA(z_i))$, where $PA$ are the parents of $z_i$, and $z_i=f_i(x_i)$ are continuous latent representations, can in principle be learned with the outlined approach. Any goal-function that can be derived from this formulation, given an appropriate latent model and an appropriate entropy maximization procedure, thus performs DM.  However, since computing entropy practically always requires some form of approximation, the precise answer to the question depends on what is still considered a "valid" approximation. The derivations in the paper make use of straightforward entropy estimators such as KNN or parameteric (e.g., Gaussian) estimators.  We found that the entropy estimators used for deriving stopgrad- and regularization-based approaches make the strongest assumptions about the latent distribution, which also affects the associated identifiability results (discussed in more detail in Figure A4). At this point we cannot exclude that it is also possible to derive other established approaches (like clustering based SSL such as SwAV) from the DM goal-function, although doing so may require more esoteric entropy approximations. We will make sure to more clearly articulate these families that we can cleanly derive and add a brief discussion regarding the general formulation of DM to the appendix.
>
> Q2: What are the strongest assumptions for identifiability?
>
> Thanks for this question. In practice, the most important assumption is that (i) "true" prediction errors are Gaussian. As discussed with Reviewer 1, the approach is robust and there is some leeway as long as the distribution remains unimodal. The assumption (ii), invertibility of f on the data manifold, arises automatically through entropy maximization. Finally, (iii) the predictor covering the latent space, we consider a mild assumption, which basically means that all dimensions in latent space have some predictive relationship with the past. Both assumptions (ii) and (iii) are, therefore, not explicitly enforced in our models. However, (iii) is enforced in the "true" system, i.e., it holds by construction of the dataset.
> We will discuss the weight of our assumptions more clearly in the updated manuscript.
>
> Q3: Can the authors give more empirical comparison showing the relation of MI maximization and DM?
>
> We agree with the reviewer that the claim that approximate MI maximization performs DM is one of the strongest in the paper and that it requires careful discussion. In our manuscript, we have given a formal argument for this correspondence, and investigated it empirically, showing that i) across several different entropy estimators and goal functions there is little difference in representation learning with or without MI on natural images (Figure 3, Table 2) and predictive SSL (Figure 4) ; ii) gradients of loss functions with or without MI maximization are consistently aligned during learning both for predictive SSL and pairwise image learning (Figure 4, A1). We now have conducted additional experiments on the SVHN dataset, showing the same results hold. We are also exploring additional approaches - for instance, showing that the encoder is invertible on the data manifold, which would imply that MI is maximized and hence $F_{MI}\propto F_{DM}$. Showing this  conclusively is challenging, but our preliminary results are consistent with the claim (cf. response to Reviewer 1). Ultimately, we likely cannot prove that approximate MI maximization performs DM in all practical scenarios. However, all the  tests we have conducted - across multiple datasets, entropy estimators, and latent geometries - support this as a reasonable conjecture. We will make sure to discuss this limitation in the final manuscript more carefully and revise our claims accordingly.

---

### Official Review · Reviewer_CJWP · 2026-03-14

**Soundness:** 3
**Presentation:** 3
**Significance:** 3
**Originality:** 3
**Overall Recommendation:** 5
**Confidence:** 4

**Summary:**

This paper introduces a theoretical framework based on distribution matching (DM) that unifies many common SSL methods (e.g. VICreg, SimCLR). This is the core contribution. The authors show how this framework can lead to new SSL methods and instantiate a particular predictive SSL model with Kalman-based latent dynamics. Finally, they provide some identifiability results for predictive SSL models that perform DM, showing that under certain assumptions optimizing the DM objective would lead to recovering the true latent variables up to trivial transformations.

**Compliance With Llm Reviewing Policy:**

Affirmed.

**Key Questions For Authors:**

The questions correspond to the weaknesses above.
A. Do you expect the the proposed Kalman-based predictor would prove useful on real high-dimensional timeseries, such as videos or finance data? It would be interesting to look into the uncertainty quantification over latent states in a practical setting.
B. Have you considered an empirical study of the degree to which entropy maximization encourages invertibility. One suggestion for this is to track the rank of the Jacobian during training, at the training points. This could be compared for models with or without the entropy term.
C. Have you investigated how much identifiability degrades under realistic violations of its assumptions? One possibility is to replace the Gaussian with a different distribution, such as a heavy-tailed one or a mixture of Gaussians, and see how much the identifiabilty degrades based on the mismatch between the learned latent distribution and the true one in the synthetic setting.

**Limitations:**

None.

**Strengths And Weaknesses:**

Strengths:
1. The unification of many SSL methods under the distribution matching framework is interesting and (somewhat) novel. Overall the paper does a good job showing that many common SSL methods can be interpreted under this framework, with appropriate choice of latent model P_theta and entropy estimator.
2. The Kalman filter model is simple, but shows that this framework could potentially be useful for deriving new SSL methods. Formulating the predictor as a Kalman filter allows principled uncertainty quantification over latent states, which is not true for standard predictive SSL.
3. The paper offers what I think is valuable insight into why stopgrad methods like BYOL and SimSiam work without contrastive objectives. The framework reveals these are implicitly performing approximate conditional entropy maximization, as the stopgrad trains the predictor to estimate the conditional distribution and uses that estimate to maximize entropy. This is a more principled explanation than earlier ones.
4. The framework offers a view of SSL that seems new. Rather than these methods working because they get rid of irrelevant information it instead suggests they preserve nearly all information on the data manifold, but obtain an advantage by geometric reparameterizing the manifold based on the choice of latent model.
5. Prior empirical work showed that MI maximization is neither necessary nor sufficient for good SSL, despite many methods stating it is a motivation. The DM framework shows that MI is already maximized when the encoder is approximately invertible, so explicit MI maximization adds little. This implies that MI-based methods work because they implicitly perform DM through constrained predictors.

Weaknesses:
1. The novelty is limited. Many SSL methods have been shown to relate directly to DM, such as contrastive SSL as conditional DM by Zimmermann et al. (2021), InfoNCE as variational inference (Aitchison & Ganev, 2024) and VICReg from an information theory view (Shwartz-Ziv et al. 2023). ICA has also been cast as a form of DM, and earlier work including the nonlinear ICA papers established the connection between DM and identifiability. This paper does however unify a wide variety of SSL methods from a DM perspective.
2. The empirical results are ok but not too compelling. The finding that DM and DM+MI perform almost equally (Table 2, Figure 3) is a useful empirical confirmation, but is consistent with what the theory predicts and does not itself establish the value of the framework.
3. Some of the key assumptions underlying the theory advanced in the paper are quite strong and may not hold in practice. The encoder is required to be invertible on the data manifold, but this is hard to verify or enforce. The paper proposes that entropy maximization encourages invertibility, but this argument is asymptotic and informal.
4. The Gaussian noise assumption in Theorem 1 is restrictive and not likely to hold in real use-cases.

---

> ### Author Rebuttal · Authors · 2026-03-31
>
> We thank the reviewer for their thoughtful comments prompting us to further improve our paper.
>
> A) (Usefulness of Kalman filter on real-world data) Thanks for raising this point.  We now include an application on a high-dimensional real-world dataset. Specifically, we ran our filtering approach on Hippocampal activity recordings of behaving rats (hc11 on CRCNS.org). Our preliminary experiments show that the Kalman style predictor learns the latent dynamics corresponding to the animal's position in the environment, which is in agreement with the non-linear encoding of space in the Hippocampus. Our preliminary experiments further show that rat positions and SSL-encoded activity are clearly organized on circular object in latent space. While this particular dataset has been investigated previously (Schneider et al. 2023), here we improve on these results by also computing confidence intervals on the decoded position, a unique feature of our approach based on the Kalman predictor (see also reviewer 4, question 2). We will include these findings in the camera ready version.
>
> B) (Does entropy encourage invertability?) Following the reviewers suggestions,  we have investigated the average local rank of the Jacobian during learning and found that upon initialization it is approximately full rank, while during learning it typically decreases to a fixed point (e.g., for CIFAR10 from rank ~512 to ~250-300, depending on the entropy estimator)---without entropy maximization it collapses. While this shows that entropy regularizes the Jacobian, we are interested in the invertibility on the data manifold, and this approach is only indirectly informative about that property. We therefore turned to synthetic datasets with a low-dimensional latent space and an invertible neural network generation function, as in many identifiability experiments (for details see reviewer 4, question 3). We then computed the Jacobian for the entire generator-encoder function, finding that it consistently is full rank throughout training. Ideally we would like to further test whether there is global invertibility---this, however, is more challenging and we continue investigating viable approaches to this. See also Reviewer 2, Q3.
>
> C) (Identifiability under realistic violations of the assumptions) This  question prompted us to conduct additional tests. Specifically, we replaced Gaussian noise with generalized Gaussian noise with short- or long-tail (e.g. Laplace) distributions. Reassuringly, identification was robust to these changes. However, extremely long-tailed noise makes learning more challenging due to the outliers, leading to noisier recovery. Finally, we tested robustness to multimodal distributions, and we found in our preliminary simulations that this is where the latent representations start to collapse. We will include and discuss these new findings and limitations in the camera ready version.

---

> > ### Author Rebuttal · Reviewer_CJWP · 2026-04-03
> >
> > I thank the authors for their response.

---

### Decision · Program_Chairs · 2026-04-30

**Decision:**

Accept (spotlight)

**Comment:**

I follow the reviewers' unanimous recommendation to accept the paper.

The authors unify several self-supervised learning methods within a new distribution matching framework and present new identifiability theory within their framework.

Reception of this work and its contributions was overall very positive, with few limitations highlighted by reviewers, e.g. limited novelty/unclear distinction from prior theoretical work, limited empirical results on larger scale datasets and benchmarks, as well as unrealistic assumptions. Some of these concerns were addressed (e.g., the distinction to i-con as a conceptually close paper, or experiments with higher dimensional data) with additional experiments and clarifications, some were out of scope for this paper, which I agree with.

When preparing the camera-ready version, the authors should carefully revisit any promises in their rebuttal comments to update the paper with experiments and clarifications and include the promised changes in the camera ready.

-SAE